# Mixture-of-Gaussian Evidential Learning for Uncertainty-Aware Stereo Matching

## Abstract

Stereo matching remains a challenging task due to the presence of uncertainties in real-world data, particularly in textureless, occluded, or reflective regions. While existing methods incorporate uncertainty estimation into stereo matching for better performance, they typically assume a pixel-wise unimodal Gaussian distribution. However, the depth distributions in real-world scenarios are rarely unimodal, making the single-Gaussian assumption inadequate for modeling their heteroscedastic and multimodal characteristics. We address this limitation with a new evidential learning framework that models each pixel with a Gaussian mixture distribution. Each mixture component is regularized by an inverse-Gamma prior, and the network predicts pseudo-posterior mixture probabilities, enabling principled per-component uncertainty estimation. We evaluate our method on stereo matching by training on the Scene Flow dataset and testing on KITTI 2015 and Middlebury 2014. Experimental results consistently show that our approach outperforms baseline methods and achieves new state-of-the-art performance on both in-domain and cross-domain benchmarks, demonstrating the robustness and effectiveness of the proposed framework. The code will be publicly released upon completion of the review process.

## 1 Introduction

Stereo matching estimates a dense disparity field from a rectified image pair, enabling depth recovery for applications such as autonomous driving, robotics, and AR/VR systems. Despite substantial progress using cost–volume convolution networks and transformer-based matchers Cao et al. (2019); Guo et al. (2019); Li et al. (2021); Lou et al. (2023); Cheng et al. (2025), achieving accurate matching in real-world environments remains challenging. In particular, textureless regions, occlusions, and reflective or transparent surfaces often lead to **ambiguous correspondences**, sharp discontinuities, and domain shifts, which in turn degrade generalization.

In safety-critical applications, obtaining reliable uncertainty estimates is therefore essential, both for risk-aware deployment and for improving the depth estimation itself Wang et al. (2025a); Gawlikowski et al. (2023). Classical Bayesian neural networks Neal (2012); MacKay (1992); Kendall & Gal (2017) can capture epistemic and aleatoric uncertainties, but they require costly sampling and careful prior specification. Evidential deep learning offers a more efficient alternative by learning the hyperparameters of a predictive likelihood in a single forward pass Gao et al. (2024), significantly reducing computational overhead. In regression tasks, deep evidential regression Amini et al. (2020a) models targets as a single Gaussian with unknown mean and variance under a Normal–Inverse-Gamma (NIG) prior, producing closed-form estimates of epistemic and aleatoric uncertainties without sampling.

However, relying on a *single* Gaussian Amini et al. (2020a) is often too restrictive for the heteroscedastic noise patterns common in stereo matching. Such a simplification fails to account for the diverse sources of noise present in real-world scenarios. For example, two pixels corresponding to the same true depth may differ significantly in difficulty due to textureless regions, occlusions, shadows, or abrupt edge changes, as illustrated in Figure 1. This motivates the need for a more flexible uncertainty modeling approach that can capture the heterogeneity inherent in stereo disparity estimation.

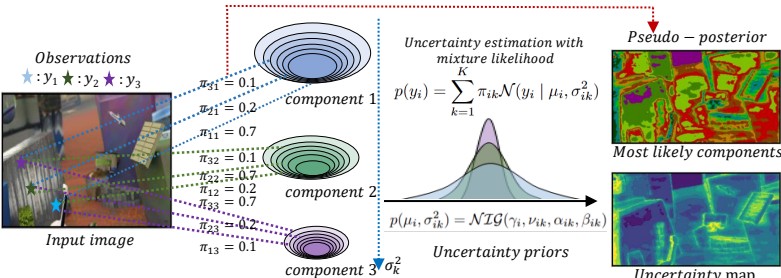

Figure 1: Each pixel's depth is modeled by a Gaussian mixture with multiple variances $\sigma_k^2$ to capture heterogeneous noise. Under a Normal–Inverse-Gamma prior and mixture likelihood, the network predicts pseudo-posterior component responsibilities $\pi_k$ and uncertainty estimates. Right: Top contributing component (top) and aggregated uncertainty map (bottom). High uncertainty concentrates at object boundaries, occlusions, and textureless regions; pixels with similar noise characteristics tend to select the same mixture component.

We propose modeling each pixel with a *mixture* of Gaussians with multiple variances, reflecting that all components correspond to the heterogeneous noise regimes. To enable principled uncertainty quantification, an NIG prior is imposed on the variance of each mixture component. Instead of explicit Bayesian updates, our network directly predicts mixture responsibilities and NIG parameters, interpreted as pseudo-posterior parameters in line with the evidential learning paradigm. This amortized inference allows efficient estimation of both aleatoric and epistemic uncertainties while capturing heterogeneous noise across pixels.

We integrate this mixture-of-Gaussian evidential learning framework with the STTR backbone Li et al. (2021) for the stereo matching task. Experiments on several public datasets show consistent improvements in accuracy and model uncertainty calibration. Our method achieves new state-of-the-art performance in both in-domain and cross-domain settings, demonstrating its effectiveness and robustness for stereo matching. The main contributions of this paper are summarized as follows:

- We propose a mixture-of-Gaussian-based deep evidential learning framework for estimating both aleatoric and epistemic uncertainties with flexible heterogeneous noise modeling. This framework facilitates precise regression predictions together with reliable uncertainty estimates, enhancing alignment with real-world scenarios.

- We conducted extensive experiments on Scene Flow, KITTI 2015, and Middlebury 2014. These experiment results consistently showed an improved accuracy with sharp boundary compared to the baseline and demonstrated calibrated model uncertainty and improved cross-domain generalization.

- We observe that uncertainty is not uniformly distributed: even pixels at the same depth form clusters with varying uncertainty levels. Boundaries, edges, and error-prone regions consistently show elevated variance, which our mixture-of-Gaussians framework interprets as evidence of heterogeneous noise regimes. The structured uncertainty pattern indicates that different mixture components capture distinct sources of noise in the depth estimation.

## 2 RELATED WORKS

### 2.1 UNCERTAINTY ESTIMATION

Uncertainty estimation is crucial in safety-critical applications such as autonomous driving. Bayesian neural networks Neal (2012); MacKay (1992); Kendall & Gal (2017) provide a principled approach by placing distributions over weights, but their high-dimensional parameter space makes inference challenging. Approximate methods such as variational dropout Molchanov et al. (2017) and Monte Carlo dropout Gal & Ghahramani (2016b;a) reduce this cost, while deep ensembles Lakshminarayanan et al. (2017b); Chitta et al. (2018); Wen et al. (2020); Lakshminarayanan et al. (2017a) improve robustness by aggregating diverse models, but incur substantial computational expense.

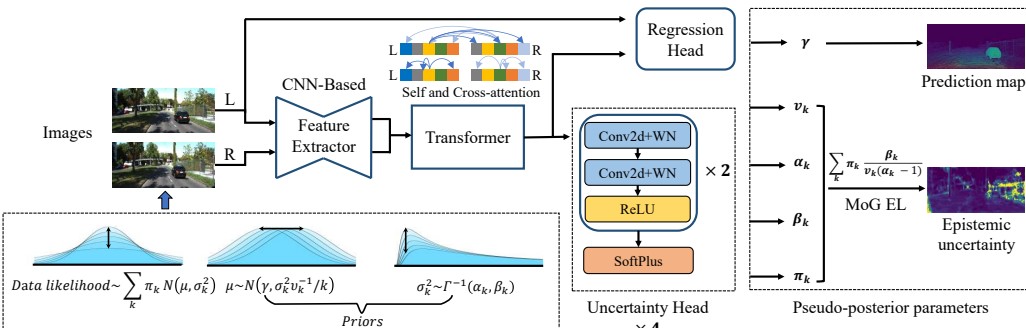

Figure 2: Overall framework of the Mixture-of-Gaussians Evidential Learning. The model produces predictions together with their epistemic uncertainties through a hierarchical probabilistic formulation. Instead of a single Gaussian likelihood, the regression target is modeled with a Gaussian mixture where all components have different variances $\sigma^2$. This enables capturing heterogeneous noise levels around the predicted mean. A Normal–Inverse-Gamma prior is further imposed over the component-level parameters $(\mu, \sigma_k^2)$, enabling principled estimation of both aleatoric and epistemic uncertainties. This hierarchical distribution effectively captures heterogeneous noise regimes. Given a set of input images, the network is trained to estimate the pseudo-posterior parameters of the hierarchical evidential distribution directly from data evidence.

To address efficiency, deterministic approaches estimate uncertainty in a single forward pass Możejko et al. (2018); Liu et al. (2020); Van Amersfoort et al. (2020); Corbière et al. (2019); Mukhoti et al. (2020). Evidential learning Sensoy et al. (2018) and prior networks Malinin & Gales (2018) introduced Dirichlet-based priors for classification, while evidential regression Amini et al. (2020a) extended this idea to continuous targets via the normal-inverse-gamma (NIG) distribution. More recently, Ma et al. (2021) extended evidential regression by predicting multiple sets of NIG parameters from different network branches and fusing them into a single equivalent NIG distribution through a parameter update rule. This approach is better understood as multi-branch NIG fusion rather than an exact probabilistic mixture model. Other recent probabilistic works include Stereo Risk Liu et al. (2024), which utilizes continuous risk minimization to improve the regression objective, and multivariate Gaussian models Liu et al. (2023) that capture spatial correlations for single-image depth prediction.

In contrast, We move beyond the single-Gaussian assumption Amini et al. (2020a); Lou et al. (2023) by introducing a mixture model with multiple variance components. This explicit mixture formulation at the data likelihood level captures heteroscedastic and multimodal noise and provides richer epistemic uncertainty estimation through pseudo-posterior responsibilities.

## 2.2 DEEP STEREO MATCHING

Stereo matching is a challenging regression task where uncertainty estimation is especially valuable for improving depth accuracy and cross-domain generalization. Early CNN models such as Disp-NetC Mayer et al. (2016) and GC-Net Kendall et al. (2017) constructed cost volumes via correlation or concatenation, with later works improving efficiency through group-wise correlation (GwcNet Guo et al. (2019)) and attention-aware volumes (ACVNet Xu et al. (2022)). Multi-scale strategies further enhanced efficiency and resolution, including hierarchical volumes (HSMNet Yang et al. (2019)), feature pyramids (AANet Xu & Zhang (2020)), and cascaded refinement (CFNet Shen et al. (2021a), PCWNet Shen et al. (2022)). With the advent of transformers Vaswani et al. (2017), STTR Li et al. (2021) and CSTR Guo et al. (2022) leveraged sequence-to-sequence matching and global context integration to handle textureless or reflective regions. However, both CNN- and transformer-based models remain vulnerable to cross-domain shifts.

## 3 METHODS

In this section, we introduce our mixture-of-Gaussians (MoG) based evidential learning framework for uncertainty estimation. As shown in Figure 2, the framework uses a baseline neural network for feature extraction, with multiple output heads predicting the parameters of a mixture-of-Gaussian and its priors in NIG in the hierarchical evidential distribution. By explicitly modeling prediction uncertainty, the framework reduces overfitting and enhances robustness for cross-domain generalization.

### 3.1 MIXTURE OF GAUSSIAN-BASED EVIDENTIAL LEARNING

To enable more flexible modeling of heterogeneous noise at the pixel level, we assume each target $y_i$ follows a Gaussian mixture distribution with $K$ distinct variances:

$$p(y_i) = \sum_{k=1}^{K} \pi_{ik} \mathcal{N}(y_i \mid \mu_i, \sigma_{ik}^2), \tag{1}$$

where $\pi_{ik}$ are non-negative mixture coefficients that satisfy $\sum_{k=1}^{K} \pi_{ik} = 1$. This formulation allows each $y_i$ to be explained by a distinct variance component.

Following the hierarchical distributions in evidential learning, we assume the variable pair $(\mu_i, \sigma_{ik}^2)$ takes the form of the NIG distribution with different hyperparameters:

$$p(\mu_i, \sigma_{ik}^2) = \mathcal{NIG}(\gamma_i, \nu_{ik}, \alpha_{ik}, \beta_{ik}) = \frac{\beta_{ik}^{\alpha_{ik}} \sqrt{\nu_{ik}}}{\Gamma(\alpha_{ik})\sqrt{2\pi\sigma_{ik}^2}} \left(\frac{1}{\sigma_{ik}^2}\right)^{\alpha_{ik}+1} exp\left\{-\frac{2\beta_{ik} + \nu_{ik}(\gamma_i - \mu_i)^2}{2\sigma_{ik}^2}\right\}. \tag{2}$$

The above prior on each component helps to model the predictive epistemic uncertainty with details given in Section 3.2.

#### 3.1.1 EVIDENCE MAXIMIZATION

Having formalized the likelihood and the prior distributions of the target variable $y_i$, we next describe our approach to estimate the hyperparameters $\mathbf{m}_i = (\pi_{ik}, \gamma_i, \nu_{ik}, \alpha_{ik}, \beta_{ik})$ for $1 \le k \le K$, which are then used to compute both predictions and epistemic uncertainty.

We adopt an evidence maximization (type-II maximum likelihood) approach, integrating out the latent mean and variance parameters. Given the likelihood in (1) and the Normal–Inverse-Gamma priors in (2), the marginal distribution of $y_i$ becomes:

$$p(y_i \mid \mathbf{m}_i) = \int_{\sigma_{ik}^2}^{\infty} \int_{\mu_i}^{\infty} \sum_k \pi_{ik} p(y_i \mid \mu_i, \sigma_{ik}^2) p(\mu_i, \sigma_{ik}^2 \mid \gamma_i, \nu_{ik}, \alpha_{ik}, \beta_{ik}) d\mu d\sigma_{ik}^2$$

$$= \sum_k \pi_k \text{St}(y_i, \gamma_i, \frac{\beta_k(1+\nu_k)}{\nu_k \alpha_k}, 2\alpha_k), \tag{3}$$

where St denotes the Student-t distribution. Detailed derivations are given in the Appendix A.2. Intuitively, this corresponds to a mixture of Student-t distributions.

The hyperparameters $\mathbf{m}_k = (\pi_k, \gamma, \nu_k, \alpha_k, \beta_k)$ are estimated by maximizing the marginal likelihood, or equivalently, minimizing the negative log-likelihood (NLL) of the observed targets $Y$:

$$\mathcal{L}^{\text{NLL}}(\mathbf{w}) = \ln\{p(Y \mid \mathbf{m})\} = \ln\left\{\prod_{i=1}^{N} \sum_{k,i} \pi_{ki} \text{St}(y_i, \gamma_i, \frac{\beta_{ki}(1+\nu_{ki})}{\nu_{ki}\alpha_{ki}}, 2\alpha_{ki})\right\}$$

$$= \sum_{i=1}^{N} \ln\left\{\sum_{k,i} \pi_{ki} \text{St}(y_i, \gamma_i, \frac{\beta_{ki}(1+\nu_{ki})}{\nu_{ki}\alpha_{ki}}, 2\alpha_{ki})\right\}, \tag{4}$$

where $1 \leq i \leq N$ indexes data points, $1 \leq k \leq K$ indexes mixture components, and typically $K \ll N$. This objective can be interpreted as a mixture-of-experts loss with evidential components, where each component contributes to both prediction and uncertainty estimation.

In evidential regression, the hyperparameters $(\alpha, \beta, \gamma, \nu)$ of the Normal–Inverse-Gamma distribution are interpreted as pseudo-posterior parameters. Although they resemble conjugate prior parameters, the network outputs them after training in an amortized Bayesian fashion: the hyperparameters already encode both prior knowledge and the evidence provided by the data, without requiring an explicit Bayesian update. In contrast, the mixture coefficients $\pi_{ik}$ obtained from directly minimizing the marginal negative log-likelihood behave only as prior mixing weights, since they are estimated independently of the observation $y_i$. The actual posterior responsibility of component $k$ for datapoint $y_i$ is instead given by

$$r_{ik} = p(z_i = k \mid y_i) \propto \pi_{ik} \, \mathrm{St}(y_i \mid \gamma_i, \nu_{ik}, \alpha_{ik}, \beta_{ik}), \tag{5}$$

which combines the prior $\pi_{ik}$ with the likelihood. This asymmetry implies that while $(\alpha_{ik}, \beta_{ik}, \gamma_i, \nu_{ik})$ can be directly used for epistemic and aleatoric uncertainty decomposition, the $\pi_{ik}$ from NLL training cannot. To overcome this, we adopt a complete-data loss formulation with amortized Expectation and Maximization (EM), where the network directly outputs $r_{ik}$ as pseudo-posterior responsibilities. This ensures that both the mixture assignment probabilities and the evidential hyperparameters represent posterior-like quantities, enabling an interpretable framework for uncertainty estimation.

### 3.1.2 EXPECTATION AND MAXIMIZATION

We introduce a latent categorical variable

$$z_i \sim \mathrm{Categorical}(\boldsymbol{\pi_i}), \quad \boldsymbol{\pi_i} = [\pi_{i1}, \ldots, \pi_{iK}],$$

indicating which component $y_i$ belongs to. The complete-data likelihood is then

$$p(Y, Z \mid \mathbf{m}) = \prod_{i=1}^{N} \prod_{k=1}^{K} \pi_{ik}^{z_{ik}} \left[ \mathrm{St}(y_i, \gamma_i, \tfrac{\beta_{ik}(1+\nu_{ik})}{\nu_{ik}\alpha_{ik}}, 2\alpha_{ik}) \right]^{z_{ik}}, \tag{6}$$

where $z_{ik}$ indicates whether target $i$ is assigned to component $k$. Taking the logarithm,

$$\ln p(Y, Z \mid \mathbf{m}) = \sum_{i=1}^{N} \sum_{k=1}^{K} z_{ik} \left[ \ln(\pi_{ik}) + \ln \left( \mathrm{St}(y_i, \gamma, \tfrac{\beta_{ik}(1+\nu_{ik})}{\nu_{ik}\alpha_{ik}}, 2\alpha_{ik}) \right) \right]. \tag{7}$$

**Expectation step:** In classical EM, one computes the posterior responsibility using

$$r_{ik} = E[z_{ik}] = p(z_{ik} \mid y_i).$$

Instead, we amortize this step by letting the network directly predict $r_{ik}$ from the input $\mathbf{X}$:

$$r_{ik} = f(\mathbf{w}, \mathbf{w}_z, \mathbf{X}), \quad i = 1, \ldots, N, ; k = 1, \ldots, K. \tag{8}$$

**Maximization step:** Given responsibilities $r_{ik}$, the expected complete-data log-likelihood becomes

$$E_z[\ln p(Y, Z \mid \mathbf{m})] = \sum_{i=1}^{N} \sum_{k=1}^{K} r_{ik} \left[ \ln(\pi_{ik}) + \ln \left( \mathrm{St}(y_i, \gamma_i, \tfrac{\beta_{ik}(1+\nu_{ik})}{\nu_{ik}\alpha_{ik}}, 2\alpha_{ik}) \right) \right]. \tag{9}$$

Rather than iterative EM updates, we adopt an amortized EM formulation where the network outputs both responsibilities $r_{ik}$ and component hyperparameters $(\gamma_i, \nu_{ik}, \alpha_{ik}, \beta_{ik})$ in a trained end-to-end way. $r_{ik}$ plays the role of a pseudo-posterior assignment, which acts as both posterior membership probability and effective mixture weight, encoding the network's learned evidence about which noise regime is active at pixel $i$. Just as $(\nu_{ik}, \alpha_{ik}, \beta_{ik})$ quantify evidence for distributional parameters of NIG for $k$-th component, $r_{ik}$ quantifies evidence for component membership. This avoids explicitly parameterizing $\pi_{ik}$, while producing posterior-like responsibilities that naturally combine prior and data evidence.

**Training objective:** The resulting loss is the negative expected complete-data log-likelihood:

$$\mathcal{L}^{NLL} = -\sum_{i=1}^{N} \sum_{k=1}^{K} r_{ik} \mathcal{L}_k = -\sum_{i=1}^{N} \sum_{k=1}^{K} r_{ik} \ln\left(\text{St}\left(y_i, \gamma_i, \frac{\beta_{ik}(1+\nu_{ik})}{\nu_{ik}\alpha_{ik}}, 2\alpha_{ik}\right)\right). \quad (10)$$

Each component loss expands to

$$\mathcal{L}_k = \frac{1}{2}\ln\left(\frac{\pi}{\nu_{ik}}\right)\alpha_{ik}\ln(\Omega_{ik}) - \left(\alpha_{ik}+\frac{1}{2}\right)\ln\left((y_i-\gamma_i)^2\nu_{ik}+\Omega_{ik}\right) - \ln\Psi_{ik}, \quad (11)$$

with $\Omega_{ik} = 2\beta_{ik}(1+\nu_{ik})$ and $\Psi_{ik} = \frac{\Gamma(\alpha_{ik})}{\Gamma(\alpha_{ik}+\frac{1}{2})}$. This formulation recovers the single-component evidential loss in Amini et al. (2020b) as a special case, while extending it to a multi-component mixture. In addition, an empirical incorrect evidence penalty $\mathcal{L}^R$ is constructed and added to the above marginal likelihood loss in Amini et al. (2020b) to reduce evidence imposed on incorrect predictions:

$$\mathcal{L}(\mathbf{w}) = \mathcal{L}^{NLL}(\mathbf{w}) + \lambda\mathcal{L}^R(\mathbf{w}), \quad (12)$$

where the coefficient $\lambda$ is a hyper-parameter to balance these two loss terms. The penalty follows:

$$\mathcal{L}^R(\mathbf{w}) = \sum r_{ik} \mid y_i - \gamma_i \mid \cdot(2\nu_{ik}+\alpha_{ik}) \quad (13)$$

with $2\nu_{ik}+\alpha_{ik}$ being the total evidence of the prior of one component $k$ in terms of "virtual-observations" for target $y_i$ in the conjugate prior interpretationAmini et al. (2020b).

**Network Implementation:** Our framework uses a base feature extractor with five evidential heads for either responsibilities $r_{ik}$ or the component parameters of the NIG ($\gamma_{ik}, \nu_{ik}, \alpha_{ik}, \beta_{ik}$). Architectural details are given in Appendix A.4 using STTR as backbone model for stereo matching. To ensure valid parameters, we apply softplus for $\nu_k, \beta_k$ and a linear activation for $\gamma$. Training follows the amortized EM formulation where the network predicts responsibilities $r_{ik}$ and updates component hyperparameters by minimizing (12) with gradient descent. The full procedure is outlined in Algorithm 1 in the Appendix A.3.

## 3.2 ALEATORIC AND EPISTEMIC UNCERTAINTY

With the hierarchical probabilistic distribution, we are now ready to derive the prediction, aleatoric and epistemic uncertainty. The target prediction i.e., $\hat{y}_i = \mathbb{E}[\mu_i]$ can be obtained by computing the following expectations given $\pi_{ik}$ and the NIG distribution:

$$\hat{y}_i = \mathbb{E}[\mu_i] = \int\int \mu_i \sum_k \pi_{ik} p(\mu_i, \sigma_{ik}^2) d\mu_i d\sigma_{ik}^2 = \gamma_i. \quad (14)$$

The aleatoric uncertainty of each component i.e., $\mathbb{E}(\sigma_{ik}^2)$ can be calculated by

$$\mathbb{E}[\sigma_{ik}^2] = \int \sigma_{ik}^2 p(\sigma_{ik}^2) d\sigma_{ik}^2 = \frac{\beta_{ik}}{\alpha_{ik}-1}, \quad (15)$$

and the epistemic uncertainty i.e., $\text{Var}(\mu)$ follows

$$\text{Var}[\mu_i] = \int\int \sum_k \pi_{ik}(\mu_i-\gamma_i)^2 p(\mu_i, \sigma_{ik}^2) d\mu_i d\sigma_{ik}^2 = \sum_k \pi_{ik}\frac{\beta_{ik}}{\nu_{ik}(\alpha_{ik}-1)}, \quad (16)$$

with $\alpha_{ik} > 2$ for all $k$. Detailed derivations can be found in the Appendix A.1. The final prediction, aleatoric and epistemic uncertainty can be obtained by replacing the unknown evidence parameters as the pseudo-posterior expectations from the network outputs $\hat{r}_{ik}, \hat{\gamma}_i, \hat{\alpha}_{ik}, \hat{\beta}_{ik}, \hat{\nu}_{ik}$.

## 4 EXPERIMENTS

In this section, we assess the effectiveness of our Mixture of Gaussian-based Evidential Learning (MOG EL) framework in stereo matching task. The stereo matching experiments were conducted on several datasets, including Scene Flow Mayer et al. (2016), & KITTI 2015 Menze et al. (2015), and Middlebury 2014 Scharstein et al. (2014).

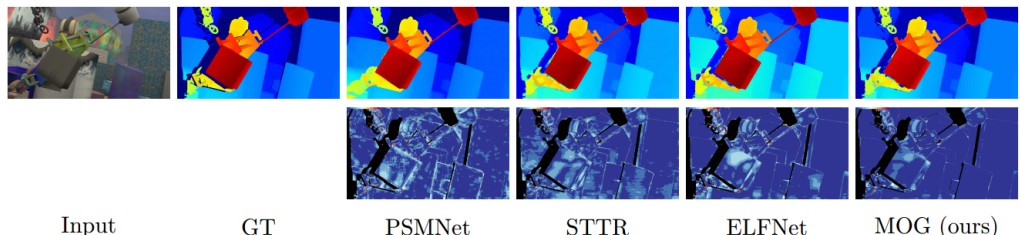

| Input | GT | PSMNet | STTR | ELFNet | MOG (ours) |

Figure 3: Results of disparity estimation for Scene Flow Flying3DThings test images with error maps. Our model show the closed results to ground truth with the minimal error.

In particular, we conducted a comparative analysis against several SOTA approaches. These include PSMNet Chang & Chen (2018), GwcNet Guo et al. (2019), CFNet Shen et al. (2021a), PCWNet Shen et al. (2022), GANet Zhang et al. (2019), STTR Li et al. (2021), CSTR Guo et al. (2022), ELFNet Lou et al. (2023), IGEV Xu et al. (2023) and MonSter Cheng et al. (2025) using different performance metics like, end-point error (EPE) and disparity outliers (D1-1px), with details given in the Appendix A.6.

## 4.1 COMPARE WITH PREVIOUS METHODS

The comparative results, as detailed in Table 1 on the Scene Flow dataset Mayer et al. (2016) with visual comparisons of predictions and the model errors are given in Figure 3. In comparison with other models, our MOG EL demonstrates superior performance by producing sharper and more accurate boundary estimations, accompanied by reduced errors in the corresponding error maps. More visual results for other samples are provided in the Appendix A.7. Our method outperforms state-of-the-art models across all evaluation metrics, including EPE and D1-1px, with particularly strong gains in scenarios covering all disparity pixels. Our research highlights the accuracy gains enabled by incorporating uncertainty estimation with a more realistic likelihood formulation. And our method establishes new state-of-the-art results on the evaluated datasets.

Table 1: Comparison with state-of-the-art on Scene Flow Mayer et al. dataset. Our method achieves a new state-of-art performance.

|  | Disparity <192 | | All Pixels | |
|  | EPE(px)↓ | D1-1px(%)↓ | EPE(px)↓ | D1-1px(%)↓ |
|---|---|---|---|---|
| PSMNet Chang & Chen | 0.95 | 2.71 | 1.25 | 3.25 |
| GwcNet Guo et al. | 0.76 | 3.75 | 3.44 | 4.65 |
| CFNet Shen et al. | 0.70 | 3.69 | 1.18 | 4.26 |
| PCWNet Shen et al. | 0.85 | 1.94 | 0.97 | 2.48 |
| GANet Zhang et al. | 0.48 | 4.02 | 0.97 | 4.89 |
| STTR Li et al. | 0.42 | 1.37 | 0.45 | 1.38 |
| CSTR Guo et al. | 0.44 | 1.41 | 0.45 | 1.39 |
| ELFNet Lou et al. | 0.43 | 1.35 | 0.44 | 1.44 |
| IGEV Xu et al. | 0.47 | - | - | - |
| MonSter Cheng et al. | 0.37 | - | - | - |
| FoundationStereo Wen et al. | 0.34 | - | - | - |
| Ours (MOG EL) | **0.31** | **1.01** | **0.33** | **1.13** |

## 4.2 MIXTURE COMPONENT ANALYSIS

Table 2: Optimal number of components of Gaussian mixture.

| #components | 5 | 10 | 20 | 50 | 100 |
|---|---|---|---|---|---|
| EPE(px) | 2.351 | 2.615 | 1.987 | 2.024 | 2.006 |

The number of components of the Gaussian mixture is found empirically by training using a subset of 1000 images. The best component number is found to be 20 as shown in Table 2, where the minimal EPE error is achieved. The illustrations of the components distribution of 4 images are

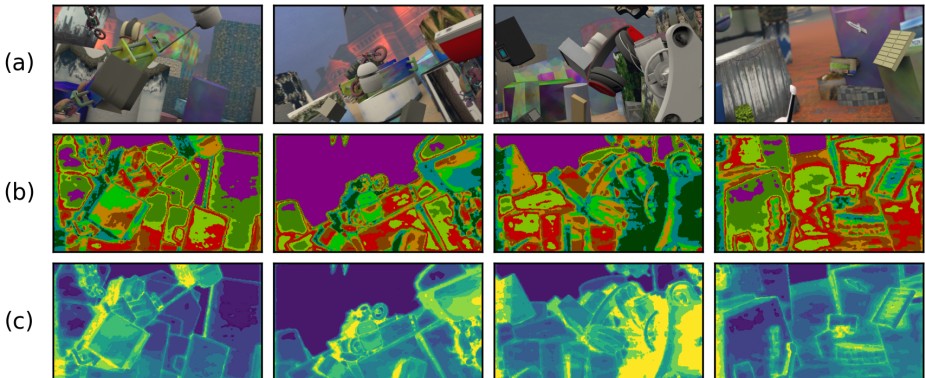

Figure 4: Components of the Gaussian mixture. (a) Original images, (b) The dominant Gaussian component for each pixel (with different components distinguished by color), (c) Corresponding epistemic uncertainty in log scale. Boundaries and high-error regions form clusters of consistently elevated uncertainty, reflecting the presence of heterogeneous noise regimes within pixels of the same depth.

given in Figure 4, where the component $k$ for each pixel $i$ is chosen with the maximum estimated pseudo-posterior probability $r_{ik}$. These components effectively segment pixels of the same depth into clusters capturing their most significant underlying variance, thereby aiding in the estimation of the depth of each object, particularly by providing clear object edges.

### 4.3 MOG EL AGAINST EL

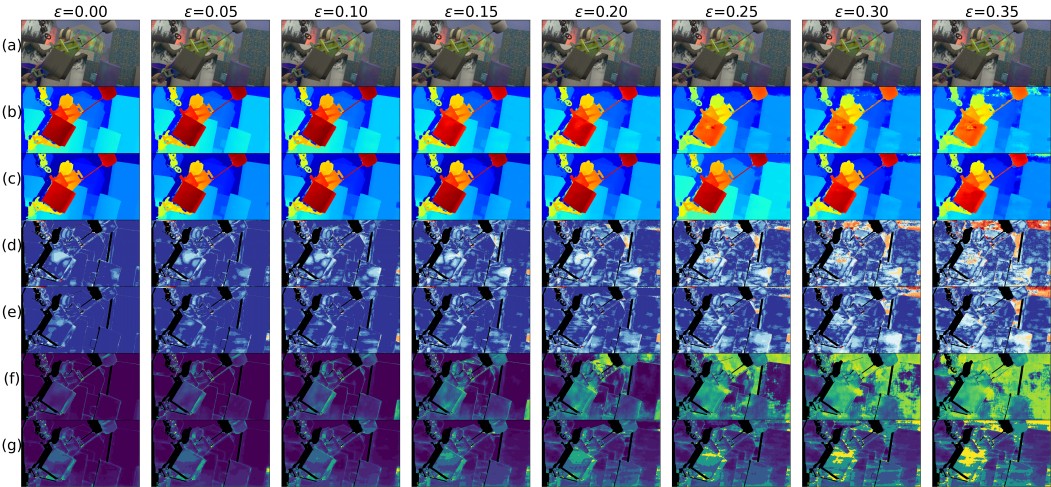

Figure 5: Effect of Adversarial Perturbation on Predictions and Uncertainty over MOG EL against EL with $\epsilon$ ranging from 0 to 0.35 with an interval of 0.05. (a) images, (b,c) predictions of EL and MOG EL, (d,e) error maps of EL and MOG EL, and (f,g) uncertainties of EL and MOG EL.

**Uncertainty against Adversarial perturbation:** To validate the robustness of the proposed MOG evidential learning framework, we conducted experiments on inputs with perturbations to challenge prediction accuracy. These perturbations were added as Gaussian noise with varying noise intensities ($\epsilon$) on Scene Flow. The impact of these adversarial perturbations on evidential uncertainty is illustrated in Figure 5, which displays the predicted depth, error, and estimated pixel-wise uncertainty. Qualitative results on the first column of Figure 5 are without additional noise. High uncertainties are observed in occluded regions, boundary regions, and areas with large errors in the error map. This result suggests that uncertainty maps provide clues for estimating accuracy. The

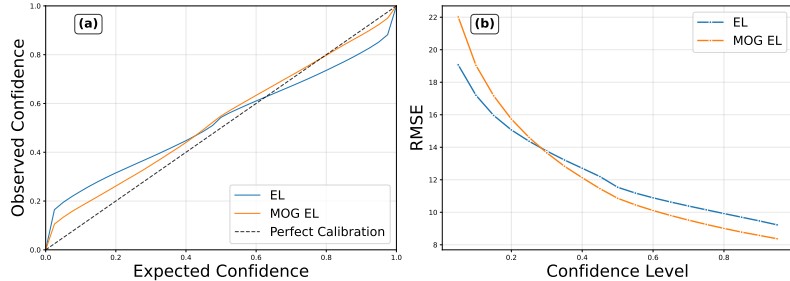

Figure 6: Comparison of Model Calibration using MOG EL and EL frameworks. (a) Calibration Curves, (b) Root Mean Square Error (RMSE) against Confidence.

results also showcase that both EL and MOG EL captures increased predictive uncertainty on samples that have been adversarily perturbed. Although ELFNet Lou et al. (2023) with conventional EL provides accurate depth prediction, it fails to capture uncertainty effectively under severe perturbations, resulting in inflated uncertainty estimates. In contrast, our model delivers more accurate depth estimations and smaller error maps as noise increases. Moreover, the epistemic uncertainty estimated by MOG EL better reflects the model error, as illustrated by the comparison between (e) and (g), even under high-perturbation scenarios.

**Performance in Model Calibration:** Figure 6(b) illustrates model performance as pixels with uncertainty above certain thresholds are progressively removed, evaluated in terms of RMSE and confidence levels. MOG EL demonstrates stronger performance than EL, with errors decreasing more steeply as confidence increases. In addition, Figure 6(a) presents the calibration of our uncertainty estimates, where calibration curves are computed following Appendix A.8 of Kuleshov et al. (2018). An ideally calibrated model aligns with the diagonal.We see that both MOG EL and EL show calibration curves close to the diagonal line. EL overestimates confidence in low confidence region and underestimates the confidence in a higher confidence region. The calibration curve for MOG EL is closer to the diagonal especially in the higher confidence scenario.

### 4.4 INTEGRATING MOG FRAMEWORK WITH OTHER BACKBONES

Table 3: **Comparison on KITTI 2015 Validation Set** with fine-tuned pretrained models. We evaluate on depth discontinuity regions using boundaries extracted from instance segmentation masks. **CE** denotes Cross Entropy loss, and **SM** stands for Single Mode inference strategy. $SEE_k$ represents the Soft Edge Error computed within a $k \times k$ local window (for $k = 3, 5$), and $\sigma(\Delta)$ indicates the percentage of outlier pixels with errors greater than $\Delta$.

| Method | $SEE_3$ | | | $SEE_5$ | | | EPE | D1-3px |
|---|---|---|---|---|---|---|---|---|
| | Avg | $\sigma(1)$ | $\sigma(2)$ | Avg | $\sigma(1)$ | $\sigma(2)$ | | |
| PSM | 1.10 | 20.57 | 9.74 | 0.99 | 17.83 | 9.02 | 0.73 | 2.49 |
| PSM + CE + SM | 1.02 | 16.12 | 7.53 | 0.90 | 13.80 | 6.94 | 0.66 | 2.09 |
| SMDNet | 0.90 | 13.09 | 6.66 | 0.79 | 10.93 | 6.01 | 0.59 | 1.95 |
| PSM + MOG (ours) | 0.44 | 8.60 | 3.05 | 0.37 | 6.95 | 2.70 | **0.58** | 1.69 |
| STTR + MOG (ours) | **0.42** | **6.49** | **2.44** | **0.32** | **4.92** | **2.16** | **0.58** | **1.51** |

To demonstrate the versatility and generalizability of our proposed method, we evaluate the Mixture-of-Gaussians (MOG) evidential head on a Convolutional Neural Network (CNN) backbone, specifically PSMNet (Chang & Chen, 2018). This contrasts with our primary implementation on the Transformer-based STTR, thereby confirming that the effectiveness of our approach is robust across fundamentally different feature extraction paradigms. The experiment serves to verify that our probabilistic formulation acts as a backbone-agnostic module that can be seamlessly integrated into various architectures to enhance uncertainty quantification. We selected PSMNet as the testbed because it serves as the foundational feature extractor for SMDNet (Tosi et al., 2021). By replacing the standard regression head of PSMNet with our MOG head, we establish a direct, "apple-to-apple" comparison with the Mixture Density head proposed in SMDNet, isolating the contribution of our evidential mixture formulation.

**Experimental Setup.** We initialized the model using pre-trained weights from the original PSMNet and fine-tuned the integrated MOG evidential head on the KITTI 2015 training set. The evaluation was conducted on the KITTI 2015 validation set. To rigorously assess the model's capability in handling challenging regions where unimodal assumptions typically failwe follow the protocol in SMDNet and report the Soft Edge Error ($SEE_k$). This metric specifically evaluates performance on depth discontinuity regions (object boundaries), which are critical for safety-aware applications. We also report the standard End-Point Error ($EPE$) and outlier rates ($\sigma$) to gauge overall regression accuracy and robustness.

**Results and Analysis.** The quantitative results are presented in Table 3. Our proposed PSM+MOG variant consistently outperforms the original PSMNet baseline, the Cross-Entropy based method (PSM + CE + SM) Chen et al. (2019), and the probabilistic SMDNet across all evaluated metrics. Most notably, in the challenging boundary regions quantified by $SEE_3$, our MOG framework achieves a substantial performance leap, reducing the Average Error from 0.90 (SMDNet) to **0.44**. This represents a reduction of over 50% in edge error, highlighting the superiority of our hierarchical evidential formulation over standard mixture density networks in modeling heterogeneous noise at discontinuities. Furthermore, our method achieves the lowest outlier rates ($\sigma$) and the best overall $EPE$ (**0.58** px) among PSM-based models. These results confirm that the MOG head effectively captures complex, multimodal noise distributions independent of the underlying feature extractor, validating its effectiveness as a general probabilistic solution.

## 4.5 Cross-domain Generalization

Table 4: **Cross-domain** evaluation **without _fine-tuning_** on Middleburry 2014 and KITTI 2015

| | Middlebury 2014 | | KITTI 2015 | |
|---|---|---|---|---|
| | EPE(px)↓ | 3px Err(%)↓ | EPE(px)↓ | D1-3px(%)↓ |
| PSMNet Chang & Chen | 3.05 | 13.0 | 6.59 | 16.3 |
| GwcNet Guo et al. | 1.89 | 8.95 | 2.21 | 12.2 |
| CFNet Shen et al. | 1.69 | 7.73 | 2.27 | 5.76 |
| PCWNet Shen et al. | 2.17 | 9.09 | 1.88 | 6.03 |
| ELFNet Lou et al. | 1.79 | 5.72 | 1.57 | 5.82 |
| STTR Li et al. | 2.33 | 6.19 | 1.50 | 6.40 |
| FoundationStereo Wen et al. | - | - | - | **4.90** |
| Ours | **1.30** | **5.49** | **1.35** | 6.52 |

To assess the impact of our mixture-of-Gaussian-based uncertainty learning on robustness, we performed cross-domain generalization experiments. A model pre-trained on the synthetic Scene Flow dataset was evaluated in a zero-shot setting on real-world benchmarks (Middlebury 2014 and KITTI 2015), with results summarized in Table 4. Compared to state-of-the-art models, our approach shows stronger generalization, achieving a notable improvement over the STTR baseline on Middlebury 2014 with EPE reduced from 2.33 to 1.3 (**44.2%**) and 3px error from 6.19 to 5.49 (11.3%). These findings highlight the effectiveness of our method in improving model robustness across domains.

## 5 Conclusion

We propose a mixture-of-Gaussians evidential learning (MOG EL) framework that extends conventional single-Gaussian and Normal–Inverse-Gamma (NIG) formation in evidential regression to a hierarchical mixture model, enabling richer modeling of heterogeneous noise and pixel-level uncertainty. Using an amortized EM, the network jointly learns pseudo-posterior responsibilities for mixture assignments and component-level NIG parameters, optimizing both mixture contributions and uncertainty estimates during training. Extensive experiments on stereo-matching task demonstrate the effectiveness of our approach. MOG-EL achieves state-of-the-art performance, delivering sharper disparity boundaries, smaller error maps, and more reliable uncertainty estimates under perturbations. Furthermore, we demonstrate that our framework functions as a plug-and-play probabilistic module, consistently yielding significant performance improvements across different backbones. Overall, the results demonstrate that hierarchical mixture-based evidential learning improves both predictive accuracy and the calibration of model uncertainty.

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

# A APPENDIX

## A.1 EXPECTATIONS

For simplicity, we omit the pixel index $i$ in the following derivations. As $\mu$ follows conditional normal distribution $\mathcal{N}(\gamma, \sigma_k^2 \nu_k^{-1})$ given the variance $\sigma_k^2$ with a probability $\pi_k$, the prediction $\hat{y} = \mathbb{E}[\mu]$ can be computed by the expectation of $\mu$ from the joint distribution:

$$\hat{y} = \mathbb{E}[\mu] = \int\int \sum_k \pi_k p(\mu, \sigma_k^2) d\mu d\sigma_k^2 \tag{17}$$

$$= \sum_k \pi_k \int\int \mu p(\mu \mid \sigma_k^2) p(\sigma_k^2) d\mu d\sigma_k^2$$

$$= \sum_k \pi_k \int p(\sigma_k^2) \int \mu p(\mu \mid \sigma_k^2) d\mu d\sigma_k^2$$

$$= \sum_k \pi_k \int p(\sigma_k^2) \langle \mu \rangle_{\mathcal{N}(\gamma, \sigma_k^2 \nu_k^{-1})} d\sigma_k^2$$

$$= \gamma \sum_k \pi_k \int p(\sigma_k^2) d\sigma_k^2$$

$$= \gamma,$$

where $\langle \mu \rangle_{\mathcal{N}(\gamma, \sigma_k^2 \nu_k^{-1})}$ denotes the expectation of $\mu$ given its distribution $\mathcal{N}(\gamma, \sigma_k^2 \nu_k^{-1})$, where $\gamma$ is shared by all $k$, $\int p(\sigma_k^2) d\sigma_k^2 = 1$ and $\sum_k \pi_k = 1$.

The data uncertainty is given by $\mathbb{E}[\sigma^2]$, which can be calculated from:

$$\mathbb{E}[(\sigma_k^2)^n] = \int (\sigma_k^2)^n p(\sigma_k^2) d\sigma_k^2 \tag{18}$$

$$= \int \frac{\beta_k^{\alpha_k}}{\Gamma(\alpha_k)} \left(\sigma_k^2\right)^{n-\alpha_k-1} exp\left\{-\frac{\beta_k}{\sigma_k^2}\right\} d\sigma_k^2$$

$$\stackrel{x := \beta/\sigma_k^2}{=} \frac{\beta_k^{\alpha_k}}{\Gamma(\alpha_k)} \int x^{\alpha_k - n - 1} e^{-\beta_k x} dx$$

$$= \frac{\beta_k^{\alpha_k}}{\Gamma(\alpha_k)} \beta_k^{n-\alpha_k} \int x^{(\alpha_k - n)-1} e^{-x} dx$$

$$= \frac{\beta_k^n \Gamma(\alpha_k - n)}{\Gamma(\alpha_k)}$$

$$= \frac{\beta_k^n}{(\alpha_k - 1)(\alpha_k - 2)\dots(\alpha_k - n)},$$

Therefore, $\mathbb{E}[\sigma_k^2] = \frac{\beta_k}{\alpha_k - 1}$, where $x := \beta_k/\sigma_k^2$ denotes to substitute $\beta_k/\sigma_k^2$ by $x$. In Amini et al. (2020a); Lou et al. (2023), the condition $\alpha_k > 1$ is imposed to ensure that the first-order statistic satisfies $\mathbb{E}[\sigma_k^2] > 0$. To achieve a more stable and reliable estimation of uncertainty, we further require the second-order statistic to be positive. Accordingly, in this work we explicitly set $\alpha_k > 2$ for all $K$ components.

Following similar procedures for the prediction estimation, the model uncertainty $\text{Var}(\mu)$ can be computed as follows:

$$\text{Var}(\mu) = \int \int \sum_k \pi_k (\mu - \gamma)^2 p(\mu, \sigma_k^2) d\mu d\sigma_k^2 \tag{19}$$

$$= \int \sum_k \pi_k p(\sigma_k^2) \langle (\mu - \gamma)^2 \rangle_{\mathcal{N}(\gamma, \sigma_k^2 \nu_k^{-1})} d\sigma_k^2$$

$$= \int \sum_k \pi_k \langle \mu^2 \rangle_{\mathcal{N}(\gamma, \sigma_k^2 \nu_k^{-1})} p(\sigma_k^2) d\sigma_k^2 - \gamma^2$$

$$= \int \sum_k \pi_k \left( \gamma^2 + \frac{\sigma_k^2}{v_k} \right) p(\sigma_k^2) d\sigma_k^2 - \gamma^2$$

$$= \sum_k \pi_k \int \frac{\sigma_k^2}{v_k} p(\sigma_k^2) d\sigma_k^2$$

$$= \sum_k \pi_k \frac{\beta_k}{\nu_k(\alpha_k - 1)},$$

where $\langle \mu^2 \rangle_{\mathcal{N}(\gamma, \sigma_k^2 \nu_k^{-1})}$ denotes the expectation of $\mu^2$ given its distribution $\mathcal{N}(\gamma, \sigma_k^2 \nu_k^{-1})$ and $\sum_k \pi_k = 1$.

## A.2 Marginal likelihood of Target variable

The marginal likelihood of the target variable $y$ given the parameters of $\gamma, \pi_k, \nu_k, \alpha_k, \beta_k$ can be achieved by integrating out $\mu$ and $\sigma_k^2$:

$$p(y \mid \gamma, \pi_k, \nu_k, \alpha_k, \beta_k)$$

$$= \int \int \sum_k \pi_k p(y \mid \mu, \sigma_k^2) p(\mu, \sigma_k^2 \mid \gamma, \nu_k, \alpha_k, \beta_k) d\mu \, d\sigma_k^2$$

$$= \sum_k \pi_k \int \int \sqrt{\frac{1}{2\pi\sigma_k^2}} \exp\left\{ -\frac{(y - \mu)^2}{2\sigma_k^2} \right\} \frac{\beta_k^{\alpha_k} \sqrt{\nu_k}}{\Gamma(\alpha_k) \sqrt{2\pi\sigma_k^2}} \left( \frac{1}{\sigma_k^2} \right)^{\alpha_k + 1} \exp\left\{ -\frac{2\beta_k + \nu_k(\gamma - \mu)^2}{2\sigma_k^2} \right\} d\mu \, d\sigma_k^2$$

$$= \sum_k \pi_k \int \frac{\beta_k^{\alpha_k} \sqrt{\nu_k} \sigma_k^{-3-2\alpha_k}}{\Gamma(\alpha_k) \sqrt{2\pi(1 + \nu_k)}} \exp\left\{ -\left( \frac{2\beta_k + \nu_k(y - \gamma)^2}{1 + \nu_k} \right) \middle/ (2\sigma_k^2) \right\} d\sigma_k^2$$

$$= \sum_k \pi_k \frac{\Gamma\left( \frac{1}{2} + \alpha_k \right)}{\Gamma(\alpha_k)} \frac{(2\beta_k(1 + \nu_k))^{\alpha_k}}{\sqrt{\nu_k/\pi}} (\nu_k(y - \gamma)^2 + 2\beta_k(1 + \nu_k))^{-\left( \frac{1}{2} + \alpha_k \right)}$$

$$= \sum_k \pi_k \text{St}(y, \gamma, \frac{\beta_k(1 + \nu_k)}{\nu_k \alpha_k}, 2\alpha_k).$$

## A.3 Pseudo-algorithm

Our training procedure follows the amortized EM formulation. The pseudo-algorithm with main steps are is outlined in Algorithm 1. The network predicts responsibilities $r_{ik}$ using one head and updates component hyperparameters by minimizing (12) with gradient descent.

## A.4 Detailed Network Layers

The detailed network structure, including specific layers, resolutions, and channels based on the STTR backbone for stereo matching, is summarized in Table 5. The "Feature Extractor" and "Transformer" constitute the STTR backbone, while the regression head performs depth regression and the remaining four heads estimate the pseudo-posterior for evidential learning. The table also shows a

---

**Algorithm 1:** Mixture-of-Gaussian based Deep Evidential Learning

---

**Input:** Dataset $\mathcal{D}$, backbone model $f(w)$

**1** **while** *not converged* **do**

**2**    **Expectation step:** Calculate posterior mixture coefficients $r_{ik}$ via Equation (8) using head $\mathbf{w}_z$ of $f(w)$;

**3**    **Maximization step:**

**4**      Estimate $\gamma_i, \nu_{ik}, \alpha_{ik}, \beta_{ik}$ via the four heads of $f(w)$;

**5**      Update $f$ by minimizing loss in Equation (12) via gradient descent;

**6** Compute prediction via Equation (14);

**7** Compute data uncertainty via Equation (15);

**8** Compute model uncertainty via Equation (16);

---

lightweight additional layers integrated into the original backbone to enable uncertainty quantification through evidential learning.

Table 5: The MOG EL network structure with STTR backbone. $H'$ and $W'$ denote the spatial dimensions of downsampled feature maps relative to the original input image. $K$ is the number of Gaussian components. In our implementation, we employ a downsampling factor of 3, where $H' = H/3$ and $W' = W/3$. Nearest neighbor upsampling is utilized for evidential visualization. Abbreviations: SPP (Spatial Pyramid Pooling), BN (Batch Normalization), WN (Weight Normalization).

| Block | Layers | Resolution | Channels |
|---|---|---|---|
| Feature Extractor | (Conv $3 \times 3$, BN, ReLU) $\times 3$ | $H/2 \times W/2$ | 16/16/32 |
| | BasicBlock $\times 3$, stride=2 | $H/4 \times W/4$ | 64 |
| | BasicBlock $\times 3$, stride=2 | $H/8 \times W/8$ | 128 |
| | SPP 1: AvgPool($16 \times 16$) + Conv $1 \times 1$ | $H/16 \times W/16$ | 32 |
| | SPP 2: AvgPool($8 \times 8$) + Conv $1 \times 1$ | $H/16 \times W/16$ | 32 |
| | SPP 3: AvgPool($4 \times 4$) + Conv $1 \times 1$ | $H/16 \times W/16$ | 32 |
| | SPP 4: AvgPool($2 \times 2$) + Conv $1 \times 1$ | $H/16 \times W/16$ | 32 |
| | SPP Concat | $H/16 \times W/16$ | 128 |
| Transformer | (Self-Attention + Cross-Attention) $\times 6$ | $H' \times W'$ | 128 |
| Regression Head: posterior parameter $\hat{\gamma}_i$ | Algorithm: Optimal Transport/Softmax | - | - |
| | Disparity Computation | $H' \times W'$ | 1 |
| | Occlusion Computation | $H' \times W'$ | 1 |
| | Nearest Neighbor Upsampling | $H \times W$ | 1 |
| Evidential Head 1: posterior responsibilities $\hat{r}_{ik}$ | Conv $3 \times 3$, WN | $H' \times W'$ | 32 |
| | Conv $3 \times 3$, WN, ReLU | $H' \times W'$ | 32 |
| | Conv $3 \times 3$, WN | $H' \times W'$ | 32 |
| | Conv $3 \times 3$, WN, ReLU | $H' \times W'$ | 32 |
| | Conv $3 \times 3$ | $H' \times W'$ | $K$ |
| Evidential Head 2: posterior parameter $\hat{\alpha}_{ik}$ | Conv $3 \times 3$, WN | $H' \times W'$ | 32 |
| | Conv $3 \times 3$, WN, ReLU | $H' \times W'$ | 32 |
| | Conv $3 \times 3$, WN | $H' \times W'$ | 32 |
| | Conv $3 \times 3$, WN, ReLU | $H' \times W'$ | 32 |
| | Conv $3 \times 3$ | $H' \times W'$ | $K$ |
| Evidential Head 3: posterior parameter $\hat{\beta}_{ik}$ | Conv $3 \times 3$, WN | $H' \times W'$ | 32 |
| | Conv $3 \times 3$, WN, ReLU | $H' \times W'$ | 32 |
| | Conv $3 \times 3$, WN | $H' \times W'$ | 32 |
| | Conv $3 \times 3$, WN, ReLU | $H' \times W'$ | 32 |
| | Conv $3 \times 3$ | $H' \times W'$ | $K$ |
| Evidential Head 4: posterior parameter $\hat{\nu}_{ik}$ | Conv $3 \times 3$, WN | $H' \times W'$ | 32 |
| | Conv $3 \times 3$, WN, ReLU | $H' \times W'$ | 32 |
| | Conv $3 \times 3$, WN | $H' \times W'$ | 32 |
| | Conv $3 \times 3$, WN, ReLU | $H' \times W'$ | 32 |
| | Conv $3 \times 3$, ReLU | $H' \times W'$ | $K$ |

## A.5 DATASETS

**Scene Flow FlyingThings 3D subset** Mayer et al. (2016), is a synthetic dataset, that provides around 25,000 stereo frames (960×540 resolution) with detailed sub-pixel ground truth disparity maps and occlusion regions.

**KITTI 2015** datasets Menze et al. (2015), sourced from real-life driving scenarios, include 200 training image pairs, with corresponding testing pairs and sparse disparity maps.

**Middlebury 2014** dataset Scharstein et al. (2014) consists of high-resolution indoor stereo pairs, from which we selected the quarter-resolution images for our experiments.

The Scene Flow dataset is accessible at `https://lmb.informatik.uni-freiburg.de/resources/datasets/SceneFlowDatasets.en.html`, the KITTI 2015 dataset at `https://www.cvlibs.net/datasets/kitti/eval_scene_flow.php?benchmark=stereo`, and the Middlebury 2014 dataset at `https://vision.middlebury.edu/stereo/data/scenes2014/`. These sources provide the necessary data for our analysis and are available for public access.

## A.6 EVALUATION METRICS

Our evaluation metrics for disparity prediction include end-point error (EPE), the percentage of disparity outliers by 1px or 3px (D1-1px / D1-3px), and the percentage of errors exceeding 3px (3px Err).

**End-Point Error (EPE)** measures the average absolute difference between predicted and ground truth disparity values:

$$\text{EPE} = \frac{1}{N} \sum_{i=1}^{N} |d_{\text{pred}}^{(i)} - d_{\text{gt}}^{(i)}| \tag{20}$$

where $d_{\text{pred}}^{(i)}$ and $d_{\text{gt}}^{(i)}$ denote the predicted and ground truth disparity at pixel $i$, respectively, and $N$ is the total number of valid pixels.

**Disparity Outlier Percentage (D1-1px / D1-3px)** computes the percentage of pixels where the absolute disparity error exceeds a specified threshold $\tau$:

$$\text{D1-}\tau\text{px} = \frac{1}{N} \sum_{i=1}^{N} \mathbb{K}(|d_{\text{pred}}^{(i)} - d_{\text{gt}}^{(i)}| > \tau) \times 100\% \tag{21}$$

where $\mathbb{K}(\cdot)$ is the indicator function and $\tau \in \{1, 3\}$ pixels.

**3-Pixel Error Rate (3px Err)** specifically measures the percentage of pixels with disparity errors greater than 3 pixels:

$$\text{3px Err} = \frac{1}{N} \sum_{i=1}^{N} \mathbb{K}(|d_{\text{pred}}^{(i)} - d_{\text{gt}}^{(i)}| > 3) \times 100\% \tag{22}$$

These evaluation metrics provide complementary perspectives on model performance. Specifically, EPE measures overall prediction accuracy with sub-pixel resolution, whereas the outlier percentages assess robustness to large deviations, which are particularly detrimental in safety-critical downstream applications such as 3D reconstruction and autonomous driving.

## A.7 MODEL PERFORMANCE COMPARISONS

Additional comparison results of disparity estimation on the Scene Flow FlyingThings3D test set using MOG-EL and other models are shown in Figure 7, including PSMNet Chang & Chen (2018), STTR Li et al. (2021), ELFNet Lou et al. (2023), and MOG. The figure presents both the predicted disparity maps and their error maps against the ground truth. As observed, MOG-EL achieves more accurate estimations in challenging regions, producing lower errors and sharper disparity boundaries.

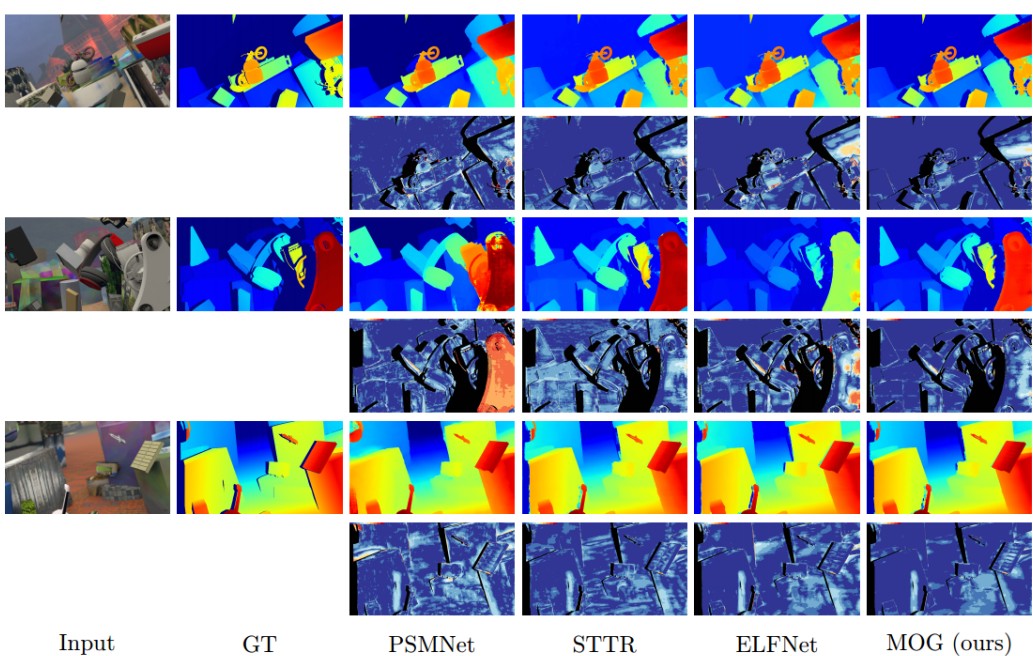

Figure 7: Results of disparity estimation for Scene Flow Flying3DThings test images. The left panel shows the input image and ground truth disparity of left image. For each input image, the disparity maps obtained by PSMNetChang & Chen (2018), STTRLi et al. (2021), ELFNetLou et al. (2023) and MOG are illustrated together above their error maps

.

## A.8 CALIBRATION CURVE

The concept of regression calibration Kuleshov et al. (2018) is formalized via predictive cumulative distribution functions (CDFs) to evaluate how well regression models are calibrated in their predictive distributions, and it has also been adopted in Amini et al. (2020a) to assess the reliability of model uncertainty quantification. The procedure involves three steps:

- Calculate the CDF for predictive probability according to

$$F(y) = P(Y \leq y \mid x) = \int_{-\infty}^{y} p(y' \mid x)dy',$$

  assuming the model outputs a predictive distribution $p(y \mid x)$ given input $x$.

- Calculate the Confidence Intervals via the CDF. For a predictive distribution, an $\alpha$-level interval is given by quantiles of the CDF according to

$$F(y_{\text{low}}) = \tfrac{1-\alpha}{2}, \quad F(y_{\text{high}}) = 1 - \tfrac{1-\alpha}{2},$$

  for different confidence levels $\alpha$, e.g., $0.1, 0.2, \ldots, 0.9$.

- Empirical coverage probability:

$$\text{Coverage}(\alpha) = \frac{1}{N} \sum_{i=1}^{N} \mathbf{1}\{y_i \in [y_{\text{low},i}, y_{\text{high},i}]\}$$

  If model is well calibrated, $\text{Coverage}(\alpha) \approx \alpha$ corresponding to the diagonal line.

## A.9 THE COMPARISON OF COMPUTATION

As shown in Table 6, we present a comparative analysis of inference time and memory usage. Compared to the baseline, our method incurs only a marginal computational overhead, primarily due to the additional heads introduced for evidential learning, as detailed in the network structure in Table 5.

| Method | FLOPs(G) | Params(M) | Test Time(s) |
|---|---|---|---|
| Baseline | 798.23 | 25.13 | 2.24 |
| + MOG | 806.79 +8.56 | 25.56 +0.43 | 2.25 +0.01 |

Table 6: Comparison of FLOPS, Params, and Test Time between our method and the baseline.

| Networks | Publication | Rainy | | Sunny | | Foggy | | Cloudy | |
|---|---|---|---|---|---|---|---|---|---|
| | | EPE ↓ | D1 ↓ | EPE ↓ | D1 ↓ | EPE ↓ | D1 ↓ | EPE ↓ | D1 ↓ |
| PSMNet | CVPR'18 | 20.86 | 50.86 | 3.67 | 27.50 | 19.56 | 58.04 | 4.44 | 30.99 |
| GwcNet | CVPR'19 | 6.21 | 48.85 | 2.96 | 23.90 | 4.72 | 43.89 | 3.76 | 29.95 |
| CasStereo | CVPR'20 | 5.01 | 33.69 | 3.61 | 22.73 | 4.14 | 31.44 | 3.86 | 26.12 |
| CFNet | CVPR'21 | 4.21 | 23.56 | 2.18 | 15.06 | 3.44 | 25.91 | 3.39 | 23.28 |
| IGEV | CVPR'23 | 1.88 | 10.96 | 1.22 | 5.08 | 1.25 | 6.58 | 1.08 | 4.20 |
| Selective-IGEV | CVPR'24 | 1.18 | 5.40 | 1.10 | 4.30 | 2.17 | 13.66 | 1.13 | 4.82 |
| MOG(ours) | - | 5.40 | 18.36 | 1.25 | 3.83 | 1.25 | 5.06 | 1.14 | 4.65 |
| LightStereo | ICRA'25 | 1.11 | 4.85 | 1.08 | 3.61 | 1.16 | 4.93 | 1.01 | 3.05 |
| MonSter | CVPR'25 | 1.15 | 5.34 | 1.03 | 3.51 | 1.15 | 5.28 | 0.99 | 3.18 |
| RobuStereo | ArXiv'25 | **0.97** | **1.94** | **0.79** | **1.49** | **0.85** | **1.61** | **0.74** | **1.35** |

Table 7: Comparison of model performance on DrivingStereo across specific weather subsets. Models above the separating line are primarily trained on the **SceneFlow** dataset. Among the models listed at the bottom, **LightStereo** and **MonSter** are trained on **mixed datasets** combining synthetic and real-world data. **RobuStereo** is trained on their **generated dataset**. All models are validated on the DrivingStereo weather subset.

## A.10 CROSS-DOMAIN ROBUSTNESS UNDER ADVERSE WEATHER CONDITION

**Experimental Setup.** Table 7 presents a comprehensive comparison between our MOG framework and other state-of-the-art methods. Crucially, the comparison groups models based on their training data to ensure a fair assessment. The models listed in the upper section (including PSMNetChang & Chen (2018), GwcNetGuo et al. (2019), CasStereoGu et al. (2020), CFNetShen et al. (2021b) , IGEVXu et al. (2023), selective-IGEVWang et al. (2024), and ours) were trained *exclusively* on the synthetic **SceneFlow** dataset and evaluated on DrivingStereo in a zero-shot manner. In contrast, the models in the lower section (LightStereoWang et al. (2025b), MonSterCheng et al. (2025), and RobuStereoWang et al. (2025c)) utilized mixed datasets combining synthetic and real-world data or specialized generated datasets during training, giving them prior exposure to real-world domain shifts.

**Results and Analysis.** As shown in Table 7, among the methods pre-trained solely on SceneFlow, our MOG model demonstrates competitive or superior robustness.

- Under **Sunny**, **Foggy**, and **Cloudy** conditions, MOG achieves state-of-the-art performance within the SceneFlow-trained group. For instance, in Foggy scenes, our EPE (1.25) matches IGEV and significantly outperforms Selective-IGEV (2.17).

- Under **Rainy** conditions, while our method performs slightly worse than IGEV, it still maintains reasonable error bounds compared to earlier baselines like PSMNet.

While models trained on mixed real-world data (e.g., LightStereo, RobuStereo) exhibit stronger overall robustness across all weather types—which is expected due to their direct exposure to real-world variations—our results highlight the effectiveness of the MOG evidential framework in handling domain shifts and atmospheric degradations without seeing any real-world weather data during training.

## A.11 ABLATION STUDY ON EMPIRICAL INCORRECT EVIDENCE PENALTY

To investigate the impact of the empirical incorrect evidence penalty term $\mathcal{L}^R$ (Eq. 13) on model uncertainty calibration, we conducted an ablation study by varying the regularization coefficient $\lambda$. This term is designed to suppress evidence inflation by penalizing misleading evidence associated with incorrect predictions.

We trained the model on the Scene Flow dataset using three different $\lambda$ values: $\{0.0, 0.02, 0.05\}$, while keeping all other hyperparameters constant. We then evaluated the calibration quality on the KITTI 2015 validation set.

Figure 8 shows that the baseline STTR model without regularization ($\lambda = 0.0$) suffers from significant over-confidence, with the calibration curve dropping well below the diagonal. This confirms the presence of evidence inflation, where the model assigns excessively high confidence even to erroneous predictions. Introducing the penalty term alleviates this issue by regularizing the evidence magnitude, shifting the calibration curves closer to the ideal diagonal. With $\lambda = 0.05$, the curve becomes slightly over-penalized in the low-confidence region but aligns much more closely with the diagonal in the high-confidence region. In contrast, a smaller value such as $\lambda = 0.02$ provides insufficient suppression and remains noticeably over-confident compared with $\lambda = 0.05$, particularly in the high-confidence regime. We therefore adopt $\lambda = 0.05$ as the default configuration, as it provides a better overall performance among the tested settings. However, Figure 8 also indicates that $\lambda = 0.05$ is still not fully optimal; values between 0.05 and 0.02 may potentially achieve even stronger alignment.

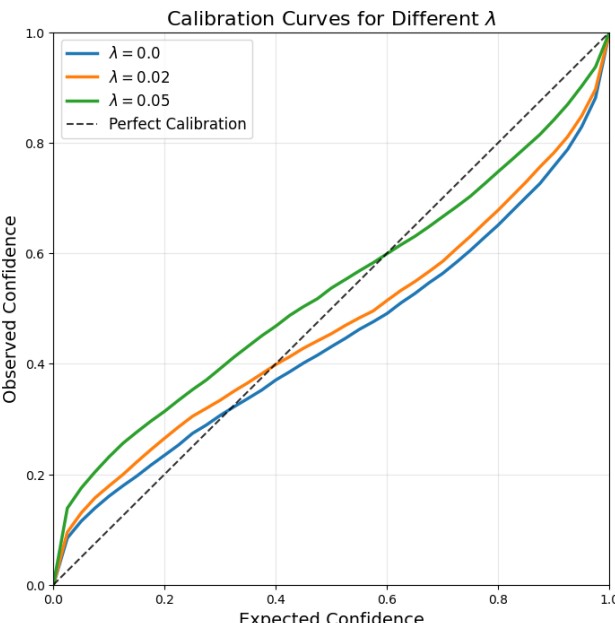

Figure 8: **Impact of the empirical incorrect evidence penalty $\lambda$ on uncertainty calibration.** The baseline model ($\lambda = 0.0$) exhibits proneness to evidence inflation, resulting in over-confident predictions. The regularization term suppresses this artifact, and $\lambda = 0.05$ yields a relatively reliable estimation, closely tracking the ideal calibration.

### A.12 ADDITIONAL QUALITATIVE VISUALIZATION RESULTS

To provide deeper insights into the working mechanism of our Mixture-of-Gaussians (MOG) framework, we present additional visualization results on real-world datasets and a detailed analysis of the learned probability distributions.

**Component Selection and Uncertainty Maps** Figures 9 and 10 illustrate the spatial distribution of the dominant Gaussian components and the corresponding epistemic uncertainty on the KITTI 2015 and Middlebury 2014 datasets, respectively.

- **Dominant Component Clustering:** As shown in row (b), the network does not select mixture components randomly. Instead, pixels belonging to similar semantic or geometric regions (e.g., road surfaces, vegetation, object boundaries) tend to select the same Gaussian component (indicated by the same color). This confirms that different components in the

mixture effectively specialize in capturing specific noise regimes inherent to different image structures.

- **Uncertainty Correlation:** Row (c) displays the epistemic uncertainty. Notably, regions with high uncertainty (yellow/bright areas) consistently align with object boundaries and textureless regions. This alignment with the component maps further validates that our MOG framework successfully identifies and models heterogeneous noise sources.

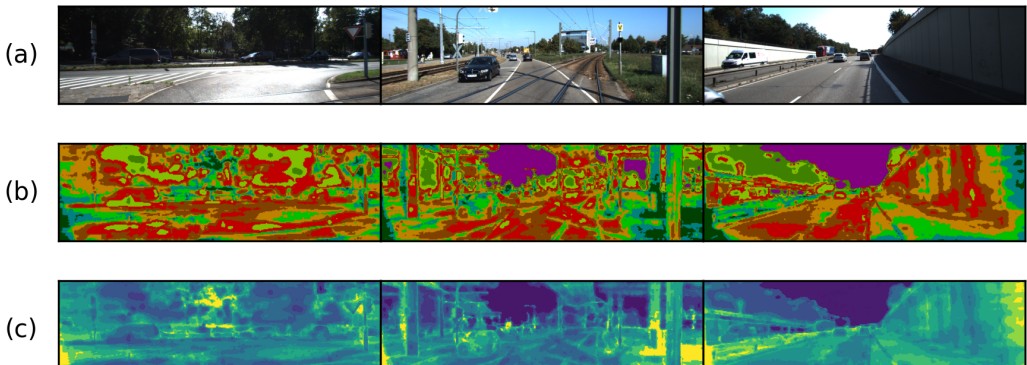

Figure 9: Visualization on **KITTI 2015**. (a) Input RGB images. (b) The dominant Gaussian component map, where different colors represent the index of the component with the highest weight ($\pi_k$) for each pixel. (c) The corresponding epistemic uncertainty map in log scale. High uncertainty concentrates at object boundaries and distant regions.

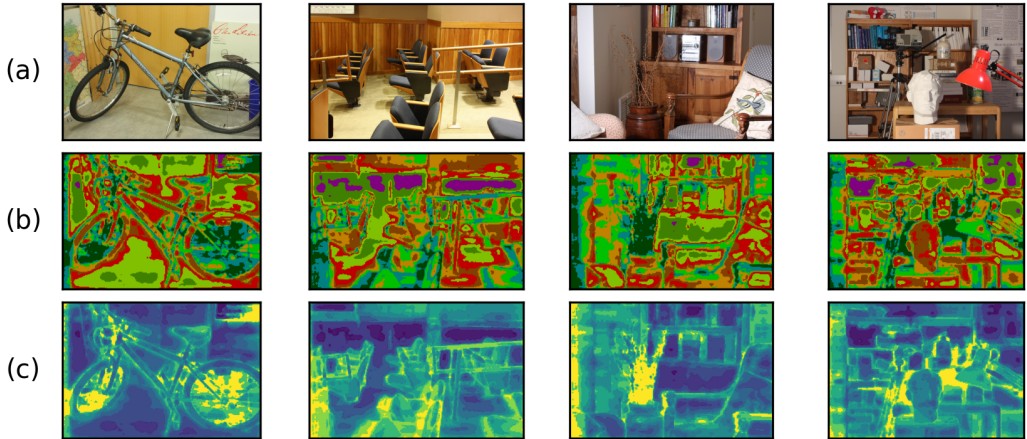

Figure 10: Visualization on **Middlebury 2014**. (a) Input RGB images. (b) Dominant Gaussian component map. (c) Epistemic uncertainty map. The structured clustering in (b) and its correlation with (c) demonstrate the model's ability to decouple heterogeneous noise regimes.

**Analysis of Learned Distributions in Different Regions** To better understand how the model handles ambiguity, we visualized the Probability Density Function (PDF) of the dominant Gaussian component for pixels in three representative regions: complex texture, textureless areas, and occlusions.

As shown in Figure 11, our framework adaptively adjusts the shape of the predictive distribution:

1. **Complex Texture (Cyan):** In regions with rich visual cues, the model predicts a sharp, low-variance distribution, indicating high confidence and precision.

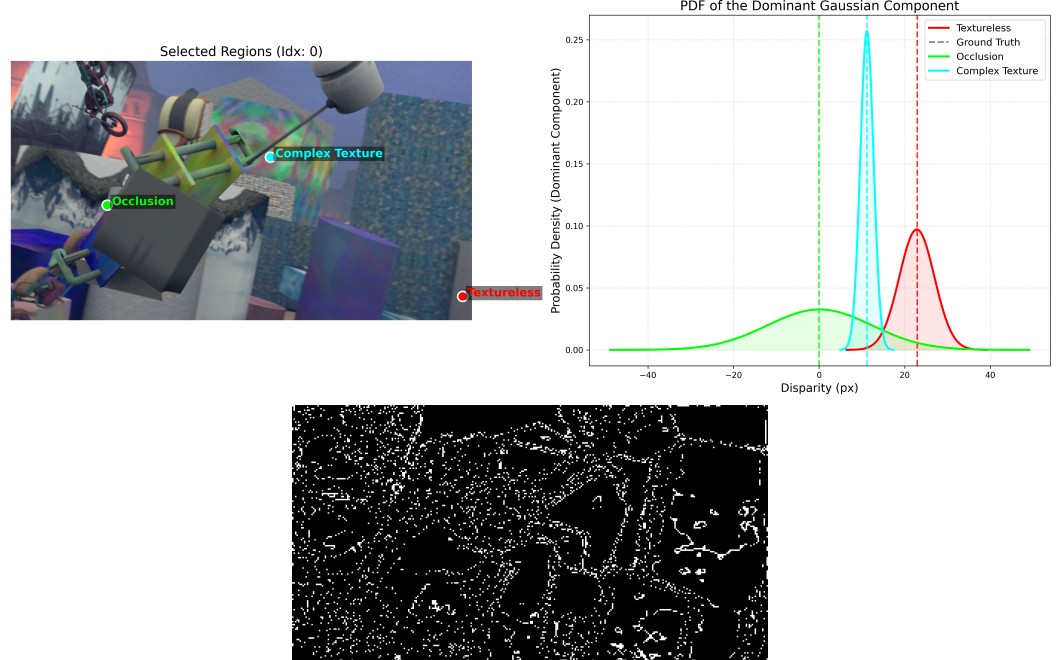

Figure 11: Visualization of the **learned Gaussian mixture distribution** for different regions. **Top Left:** Selected query points on the input image: Complex Texture (Cyan), Textureless (Red), and Occlusion (Green). **Top Right:** The PDF of the dominant Gaussian component for each point. The dashed lines indicate the Ground Truth disparity. The model adaptively outputs sharp peaks for confident regions and broad, heavy-tailed distributions for uncertain or occlusion regions. **Bottom:** Entropy mask of the Gaussian mixture weights. Pixels with normalized entropy $\bar{\mathcal{H}} > 0.2$ are visualized. The strong correlation between high-entropy regions and object boundaries confirms that our MOG framework correctly identifies areas with high matching ambiguity, assigning them a broader, multi-component probability distribution.

2. **Textureless Area (Red):** For featureless surfaces, the distribution becomes broader. This flattened shape correctly reflects the higher aleatoric uncertainty due to matching ambiguity.

3. **Occlusion/Edge (Green):** At disparity discontinuities, the distribution exhibits a significantly wider base (high variance). This heavy-tailed behavior captures the heterogeneous noise and the potential risk of outliers characteristic of edge regions.

Furthermore, we investigate the internal decision-making process of the MOG model by calculating the **normalized Shannon Entropy** $\bar{\mathcal{H}}$ of the posterior mixture probability $\boldsymbol{r}$ for each pixel. Given the predicted mixture weights $\boldsymbol{r}_i = \{r_{i1}, \ldots, r_{iK}\}$ for the pixel $i$, the metric is defined as:

$$\bar{\mathcal{H}}(\boldsymbol{r}_i) = \frac{-\sum_{k=1}^{K} r_{ik} \ln(r_{ik})}{\ln(K)} \tag{23}$$

where $K$ is the total number of Gaussian components (20 in our experiments). The value of $\bar{\mathcal{H}}$ ranges from 0 to 1. A lower $\bar{\mathcal{H}}$ indicates that the model confidently selects a single dominant component (sparse selection), whereas a higher $\bar{\mathcal{H}}$ implies that the probability mass is distributed across multiple components, reflecting high ambiguity or multimodal uncertainty. As illustrated in Figure 11, we visualize pixels with $\bar{\mathcal{H}} > 0.2$. The results reveal a strong spatial correlation: high-entropy pixels are predominantly concentrated at object boundaries and depth discontinuities. This confirms that our framework adaptively activates multiple Gaussian components to model the complex, heterogeneous noise regimes inherent in edge regions, while maintaining deterministic, single-component predictions in texture-rich, planar areas.

## A.13 NOTATIONS

As shown in Table 8, we summarize the main notations used in this paper for easy reference.

Table 8: Summary of main notations used in this paper. Here $i$ denotes pixel index, and $k$ denotes mixture component index.

| Symbol | Description |
|---|---|
| $y_i$ | Observed disparity (ground-truth) at pixel $i$ |
| $\mu_i$ | Latent disparity (true mean) at pixel $i$ |
| $\gamma_i$ | Predicted mean parameter (location) from the network |
| $\sigma_{ik}^2$ | Variance of component $k$ at pixel $i$ (heteroscedastic noise) |
| $\pi_{ik}$ | Mixture prior coefficient (prior weight of component $k$ at pixel $i$) |
| $r_{ik}$ | Posterior responsibility of component $k$ at pixel $i$ (network output) |
| $\nu_{ik}$ | Scale parameter of NIG prior (precision of $\mu_i$) |
| $\alpha_{ik}$ | Shape parameter of NIG prior (controls variance distribution, $\alpha > 2$) |
| $\beta_{ik}$ | Rate parameter of NIG prior (controls variance scale) |
| $\mathrm{St}(\cdot)$ | Student-$t$ distribution (marginal likelihood from NIG) |
| $\mathbb{E}[\sigma_{ik}^2]$ | Aleatoric uncertainty (data-dependent noise) |
| $\mathrm{Var}[\mu_i]$ | Epistemic uncertainty (model uncertainty) |

