# OpenReview forum: "Mixture-of-Gaussian Evidential Learning for Uncertainty-Aware Stereo Matching"
_ICLR.cc/2026/Conference — ICLR 2026 Conference Desk Rejected Submission_

### Official Review · Reviewer_eeUX · 2025-10-30

**Soundness:** 2
**Presentation:** 2
**Contribution:** 2
**Rating:** 4
**Confidence:** 5

**Summary:**

This paper introduces an uncertainty-aware stereo matching framework. Here, each pixel's disparity is modeled with a mixture of Gaussians, and each component is governed by a Normal Inverse Gamma prior for uncertainty estimation. The network is trained in an evidential learning paradigm rather than sampling or iterative Bayesian updates. It predicts the parameters of this mixture model, including per-component uncertainty hyperparameters and pseudo-posterior responsibilities. The authors integrate their approach with the STTR stereo transformer architecture, showing improved disparity accuracy and calibrated uncertainty estimates compared to baseline methods. Extensive experiments on Scene Flow, KITTI 2015, and Middlebury 2014 show the proposed approach's performance compared to competing methods.

**Strengths:**

1. The training approach is grounded in Expectation-Maximization (EM) formulation, but the authors avoid the complexity of iterative EM by adopting an amortized EM strategy. The network has dedicated output heads that directly predict the pseudo-posterior responsibilities for each mixture component, alongside the NIG hyperparameters per component.

2. Empirically, the proposed model shows sharper disparity boundaries and lower error rates than the baseline, especially in challenging regions.

3. This work extends evidential learning to a mixture, i.e., the approach could separate aleatoric uncertainty (expected data noise per component) and epistemic uncertainty (variance in the estimated mean across components) for each pixel. To this end, the paper provides formulae for computing per-component aleatoric variance and overall epistemic variance from the predicted NIG parameters.

**Weaknesses:**

1. A critical limitation of the proposed method is the approach model assumption that all mixture components share the same predicted mean for a given pixel disparity. In the formulation, the network predicts a single $\gamma_i$ per pixel (the mean of the latent true disparity), while the Gaussian mixture components provide different variance scales around that mean. This effectively means the mixture captures heteroscedastic uncertainty (varying noise levels) but not multiple distinct depth hypotheses for the same pixel. Kindly clarify.

2. Apart from the mixture size, other design choices are not thoroughly ablated or justified. For example, the authors include an "incorrect evidence" regularization term (analogous to Amini et al. approach) with a hyperparameter $\lambda$ to penalize predictions that have high confidence but large error. However, the paper does not detail how $\lambda$ was chosen or how sensitive the results are to this value.

3. The concept of pseudo-posterior responsibilities and how they implicitly combine prior and data is a bit hard to understand. By not explicitly predicting the prior mixture weights $\pi_k$, the network's $r_{ik}$ output is expected to encode both the prior belief and the likelihood evidence for component $k$. Nevertheless, it is not obvious to the reader what prior is assumed for the mixture weights.

4. Why are the mixture components segmented by depth (as observed in Figure 4)? The paper notes this empirical finding but doesn't delve into why the model chooses to align components with depth layers or edges. Is it because disparities at similar depths share similar uncertainty patterns (possibly due to similar local textures or occlusion states)? Such a discussion to understand the model's behavior is missing.

5. The ideas presented in the paper share quite a similar motivation to that presented in Liu et al.'s icml 2024 paper on stereo risk and Liu et al.'s cvpr 2023 paper on multivariate Gaussian. Missing such a discussion with recent papers with a similar motivation. Kindly point out the difference between this work and theirs.

**Questions:**

Q1. How are the mixture responsibilities ($r_{ik}$) handled in terms of prior assumptions? Since the network directly outputs posterior-like responsibilities without an explicit $\pi_k$, did you effectively assume a uniform prior over mixture components?

Q2. Can you clarify the interpretation of mixture weights as "pseudo-posterior" and how prior evidence is reflected? The network's output $r_{ik}$ combines prior and likelihood evidence – does this mean that when the model is unsure (e.g., in very ambiguous pixels), the responsibilities default to a roughly uniform distribution (as a prior would). In contrast, in confident cases, one $r_{ik}$ is near 1? Detailing an example of how $r_{ik}$ behaves in practice would help.

Q3. How the current model behaves compared to the current Foundation model for zero-shot stereo matching, i.e., FoundationStereo cvpr 2025.

---

> ### Author Response · Authors · 2025-11-20
> **Responses to Comments from Reviewer eeUX**
>
> • On the shared mean assumption and the inability to model multiple depth hypotheses:
> Our mixture-of-Gaussians evidential formulation deliberately enforces a shared mean across components because each pixel corresponds to a single ground-truth disparity, and introducing multiple means creates an identifiability issue under amortized EM. In our experiments, a multiple-mean variant often failed to converge or converged to spurious multi-modal predictions that did not correspond to any physically meaningful depth hypotheses. Even when convergence occurred, its performance was worse than both the shared-mean mixture and the single-Gaussian baseline. The shared-mean design resolves ambiguity by anchoring all components to a single predicted disparity while allowing their variances to specialize to heterogeneous noise regimes, yielding sharper boundaries and better-calibrated uncertainty. We note that standard iterative EM may mitigate mean-ambiguity and plan to explore this in future work, but current results consistently favor the shared-mean formulation for stability.
>
>
> • On the choice and sensitivity of the incorrect-evidence regularization term:
> We adopt the same incorrect-evidence penalty as Amini et al. and set its weighting hyperparameter to 0.05 for simplicity, since this term is not the focus of our contribution. While this fixed choice works well empirically and avoids introducing additional hyperparameters, we agree that a more thorough sensitivity analysis would strengthen the paper. We will include experiments in the revision that vary this hyperparameter and report its effect on calibration and prediction accuracy.
>
>
> • On the interpretation of pseudo-posterior responsibilities and prior assumptions:
> In our amortized EM formulation, the network directly outputs the responsibilities ( r_{ik} ), which function as posterior-like mixture assignments. Since we never parameterize or optimize explicit mixture weights ( \pi_{ik} ), the model implicitly adopts a non-informative prior over component assignments. This is analogous to a standard probabilistic GMM where the mixture weights are unconstrained beyond the simplex—mathematically equivalent to imposing a uniform Dirichlet prior with all concentration parameters equal to 1. This design avoids incorrectly interpreting ( \pi_{ik} ) as posterior responsibilities (which would be unsuitable for uncertainty decomposition) and ensures that the learned responsibilities are fully data-driven.
>
>
> • On why mixture components appear segmented by depth in Figure 4:
> We appreciate the reviewer’s observation. Empirically, components tend to specialize along depth planes because pixels at similar depths often share similar uncertainty patterns—e.g., similar textures, consistency constraints, occlusion likelihood, or sensitivity to disparity discontinuities. This leads components to naturally align with coherent depth or edge structures. Edge and high-error regions form distinct clusters with elevated variance, reflecting heterogeneous noise regimes within a shared predicted depth. We expand this discussion in the paper to clarify the underlying behavior.
>
>
> • On the relation to Liu et al. (ICML 2024) and Liu et al. (CVPR 2023):
> We  added a comparison section in the revised paper to clarify these conceptual and methodological differences. Moreover, extending our univariate Gaussian components to multivariate Gaussians with full covariance matrices and conjugate inverse-Wishart priors could be promising.
>
>
> • Q1: Prior assumption over mixture responsibilities:
> Yes. Refer to response 4. The model implicitly adopts a non-informative prior over component assignments.
>
>
> • Q2: Interpretation of responsibilities and behavior under uncertainty:
> The pseudo-posterior interpretation implies that in confident regions, the network assigns a dominant responsibility to one component, whereas in ambiguous or corrupted regions, the responsibilities become relatively diffuse and approach to a component distribution with higher entropy. We included the histogram of the entropy of the mixture distributions and explicit examples in the revision showing how posterior responsibilities concentrate for well-structured areas but relatively flatten for occlusions, or edge area.
>
>
> • Q3: Comparison with FoundationStereo (CVPR 2025):
> Experiments comparing our method with FoundationStereo are currently underway. We have incorporated the reported results of FoundationStereo into the revised paper and will add our own remaining quantitative results. The updated Table 1 shows that our method achieves better EPE on Scene Flow (0.31 vs. 0.34 for disparity < 192) compared to FoundationStereo. On KITTI 2015 in Table 4 for cross-domain evaluation, our current results are slightly worse (6.5 vs 4.9 for D1-3p); however, our model has not yet been fine-tuned on KITTI. We will include the fine-tuned KITTI 2015 results in the next revision.

---

> ### Author Response · Authors · 2025-12-02
> **Responses to analysis of the incorrect-evidence regularization term**
>
> An ablation study on the regularization coefficient for the empirical incorrect-evidence penalty (as suggested by reviewer eeUX) is provided in Appendix A.11. In this experiment, we vary the regularization strength λ, which is designed to suppress evidence inflation by penalizing misleading evidence associated with incorrect predictions.
>
> We trained the model on the Scene Flow dataset using different values of λ from {0.0, 0.02, 0.05}, while keeping all other hyperparameters fixed. Calibration quality was then assessed on the KITTI 2015 validation set, with the resulting calibration curves shown in Figure 8 of the Appendix A.11
>
> Figure 8 s shows that the baseline STTR model without regularization (λ = 0.0) suffers from significant over-confidence, with the calibration curve dropping well below the diagonal. This confirms the presence of evidence inflation, where the model assigns excessively high confidence even to erroneous predictions. Introducing the penalty term alleviates this issue by regularizing the evidence magnitude, shifting the calibration curves closer to the ideal diagonal.
>
> With λ = 0.05, the curve becomes slightly over-penalized in the low-confidence region but aligns much more closely with the diagonal in the high-confidence region. In contrast, a smaller value such as λ = 0.02 provides insufficient suppression and remains noticeably over-confident compared with λ = 0.05, particularly in the high-confidence regime. We therefore adopt λ =0.05 as the default configuration, as it provides a better overall performance among the tested settings. However, Figure 8 also indicates that λ = 0.05 is still not optimal; some values between 0.05 and 0.02 may potentially achieve stronger alignment.

---

### Official Review · Reviewer_KYM6 · 2025-10-31

**Soundness:** 2
**Presentation:** 3
**Contribution:** 2
**Rating:** 4
**Confidence:** 2

**Summary:**

This paper introduces a Mixture-of-Gaussian Evidential Learning (MOG EL) framework for stereo matching with uncertainty. It models depth using a Gaussian mixture, regularized by an Inverse-Gamma prior. The network predicts parameters to handle complex noise in real-world scenarios. Tests on the Scene Flow, KITTI 2015, and Middlebury 2014 datasets demonstrate top performance in disparity accuracy and uncertainty, especially in challenging regions like boundaries and occlusions. The paper also includes robustness tests, such as component analysis and adversarial perturbation evaluations.

**Strengths:**

1. The paper is well-written and clearly motivated, making it easy to understand the author's approach and reasoning.
2. The method achieves state-of-the-art (SOTA) performance on the Scene Flow dataset, with visual comparison results outperforming existing methods.
3. The mathematical and theoretical analysis of the proposed method is thorough.
4. The paper provides a comprehensive discussion of existing related work.
5. The motivation and framework diagrams effectively showcase the author's innovations and improvements.

**Weaknesses:**

1.The experiments are limited, with testing conducted only on the Scene Flow dataset, and only cross-domain evaluation on the Middlebury 2014 and KITTI 2015 datasets. It is recommended to include results from at least one mainstream online benchmark, such as Middlebury, KITTI, or ETH3D.
2.The ablation study lacks direct comparisons with baseline models and the single-Gaussian method. It does not clearly demonstrate that the core innovation is the primary source of performance improvement.
3.As seen in Figure 6(b), when the model's confidence is low, the proposed method shows larger errors than the EL framework method. This issue is not sufficiently analyzed.
4.In Table 2, the optimal number of components for the Gaussian mixture reaches its best performance around 20, and the performance remains stable around 2 as the number increases. Is this phenomenon related to the dataset? A deeper analysis of this experimental result is missing.

**Questions:**

Please answer my questions in the weakness part.

---

> ### Author Response · Authors · 2025-11-20
> **Responses to Comments from Reviewer KYM6**
>
> 1. Limited experiments / lack of mainstream benchmark results:
> We thank the reviewer for the suggestion. To demonstrate that our evidential mixture formulation is general and not tied to a specific architecture (STTR), we have already integrated our evidential heads into PSMNet by replacing its SDM head. Experiments on fine-tuning on KITTI and additional benchmarks are currently in progress.
> We will include these new results in the revised manuscript to provide a more complete evaluation across mainstream datasets.
> 2. Ablation study lacks comparisons with single-Gaussian baselines:
> We respectfully clarify that the comparison against the single-Gaussian (EL) baseline is already conducted in Section 4.3 “MoG-EL against EL”, where EL refers to the single-Gaussian evidential model using the same backbone.
> Figure 5 and Figure 6 present
> •	Direct performance comparison (Fig. 5, ε = 0): our MoG-EL yields noticeably lower disparity errors than its single-Gaussian counterpart.
> •	Robustness to adversarial perturbations: MoG-EL consistently outperforms EL across varying perturbation strengths.
> •	Model calibration: MoG-EL exhibits significantly better alignment between predicted uncertainty and actual error.
> Because all comparisons use the same backbone, the performance gains arise solely from our probabilistic modeling design. These results therefore demonstrate that the proposed innovation is the main driver of improvement.
> 3. Larger error in low-confidence regions (Fig. 6b):
> This behavior is expected and, in fact, desirable. A well-calibrated probabilistic model should exhibit higher error in regions where it expresses low confidence; otherwise, the model is overconfident. The higher RMSE of MoG-EL in low-confidence bins does not indicate worse overall performance. Instead, it shows that the model’s uncertainty meaningfully correlates with its prediction error—an essential property of good calibration. Similar results can be found in “Deep Evidential Regression” in Nips 2020.
> For direct performance comparison, the error maps in Figure 5 clearly show that MoG-EL achieves lower disparity error than EL across the image, confirming that overall accuracy is improved.
> 4. Behavior of performance vs. number of mixture components:
> We agree that the near-flat performance curve after 20 components. This pattern is consistent with mixture-of-Gaussians modeling: once the number of components exceeds the intrinsic complexity of the noise distribution, additional components provide little benefit. In our stereo setting, the underlying noise patterns are limited, so using more than 20 components leads to diminishing returns. As a result, performance stabilizes, and slight fluctuations around a score of 2 are expected due to optimization stochasticity. This dataset-dependent saturation is aligned with standard mixture-modeling behavior, and our empirical results follow this trend.

---

> ### Author Response · Authors · 2025-11-27
> **Response to the experiment using KITTI**
>
> To demonstrate that our method is a general probabilistic head that can be attached to different backbones, we evaluated our mixture evidential head on PSM-Net and using the data KITTI for evaluation. We chose PSM-Net because it is the backbone used in SMDNet, allowing a simple and clear apple-to-apple comparison between our probabilistic head and the SDM head.
>
> The results on the KITTI 2015 validation set (using fine-tuned pretrained models) are provided below.
> | Method               | SEE_3 Avg | SEE_3 σ>1 | SEE_3 σ>2 | SEE_5 Avg | SEE_5 σ>1 | SEE_5 σ>2 | EPE Avg | EPE σ>3 |
> | -------------------- | --------- | --------- | --------- | --------- | --------- | --------- | ------- | -------- |
> | PSM[1]                  | 1.10      | 20.57     | 9.74      | 0.99      | 17.83     | 9.02      | 0.73    | 2.49     |
> | PSM + CE + SM[2]        | 1.02      | 16.12     | 7.53      | 0.90      | 13.80     | 6.94      | 0.66    | 2.09     |
> | SMDNet [3]              | 0.90      | 13.09     | 6.66      | 0.79      | 10.93     | 6.01      | 0.59    | 1.95     |
> | **PSM + MOG (ours)** | **0.44**  | **8.60**  | **3.05**  | **0.37**  | **6.95**  | **2.70**  | **0.58**| **1.69** |
>
>
> Our MOG head consistently outperforms the SDM head and other variants across all edge-focused SEE metrics and achieves the best EPE among the PSM-based models, demonstrating the effectiveness and backbone-agnostic nature of our probabilistic formulation. The results on test set are in progress.
>
> [1] PSM: Jia-Ren Chang and Yong-Sheng Chen. Pyramid stereo matching network. In Proc. IEEE Conf. on Computer Vision and Pattern Recognition (CVPR), 2018
>
> [2] PSM + CE + SM: Chuangrong Chen, Xiaozhi Chen, and Hui Cheng. On the over-smoothing problem of cnn based disparity estimation. In Proc. of the IEEE International Conf. on Computer Vision (ICCV), 2019
>
> [3] SMDNet: Tosi et al. SMD-Nets: Stereo Mixture Density Networks. CVPR 2021

---

### Official Review · Reviewer_q5jT · 2025-11-01

**Soundness:** 3
**Presentation:** 2
**Contribution:** 2
**Rating:** 4
**Confidence:** 4

**Summary:**

The paper proposes a Mixture-of-Gaussians (MoG) evidential learning framework for uncertainty-aware stereo matching. Instead of the common single-Gaussian likelihood per pixel, each pixel’s disparity is modeled as a Gaussian mixture whose components have different variances to capture heterogeneous noise patterns (boundaries, occlusions, textureless regions). A Normal–Inverse-Gamma (NIG) prior is imposed on each component, and the network predicts component responsibilities and NIG parameters in a single forward pass, yielding estimates of both aleatoric and epistemic uncertainty. The method is plugged into an STTR-style stereo backbone and evaluated on Scene Flow (train) with testing on KITTI and Middlebury, reporting improved accuracy and better-calibrated uncertainty, including cross-domain generalization.

**Strengths:**

- A mixture likelihood modeling matches depth ambiguity at edges, occlusions, and low-texture. The qualitative observation that uncertainty clusters around such regions is well-motivated for stereo matching pipelines.
- Predicting NIG hyper-parameters per component provides closed-form uncertainty, which is practical for dense stereo.

**Weaknesses:**

- Unclear justification for the inverse-Gamma prior: The paper adopts an inverse-Gamma prior for variance following standard evidential regression but does not theoretically motivate why this choice appears to be appropriate in the proposed hierarchical mixture setting. In a mixture model, other priors (e.g., log-normal or inverse-Wishart for covariance) could better capture multi-component variance behaviors. The inverse-Gamma assumption is used without proper justification.
- Ambiguous role of hierarchical structure: The hierarchical distributions are introduced as an extension of evidential learning but their necessity is not clearly established. It is not evident how the hierarchy fundamentally improves inference beyond increasing model capacity.
- Insufficient evaluation: Although the method shows clear improvements when applied to the STTR backbone, the experimental section lacks comparisons with other stereo baselines. Additional evaluations using diverse architectures (e.g., cost-volume-based, CNN, or transformer-based models) are strongly needed to demonstrate the general applicability of the approach.

**Questions:**

Beyond the points raised in the Weaknesses, the following questions should also be addressed to further clarify the contribution.
- Cross-domain robustness: Beyond KITTI and Middlebury, how does the method address synthetic-to-real nighttime or adverse weather stereo? Any failure analyses where component variances collapse or responsibilities become diffuse?
- Training stability: Any observed mode collapse among mixture components?

---

> ### Author Response · Authors · 2025-11-20
> **Responses to Comments from reviewer q5jT**
>
> • Regarding the choice of the inverse-Gamma prior:
> We appreciate the reviewer’s comment. Our use of the inverse-Gamma prior follows directly from the standard evidential regression formulation, where each mixture component models a univariate Gaussian likelihood and the inverse-Gamma serves as its conjugate prior. This preserves a fully closed-form hierarchical structure, enabling analytic computation of aleatoric and epistemic uncertainty without relying on sampling-based or variational approximations.
> While richer priors such as log-normal are in principle possible, they introduce higher complexity and generally break conjugacy when applied to univariate likelihoods with unknown variance. The inverse-Wishart prior becomes relevant primarily when modeling multivariate Gaussian likelihoods with full covariance matrices—an extension that is orthogonal to our current formulation, which predicts scalar disparities. Because our contribution focuses on extending evidential learning from a single Gaussian to a mixture-of-Gaussians to capture noise heterogeneity, the univariate inverse-Gamma prior remains the most theoretically consistent and computationally appropriate choice.
> That said, we agree that combining mixture modeling with more expressive probability distributions—such as multivariate Gaussians with full covariance and an inverse-Wishart hierarchical prior—offers an interesting and powerful future direction. Such an extension would allow modeling structured or correlated uncertainty in stereo, and we plan to explore this in future work.
>
>
> • Regarding the role and necessity of the hierarchical structure:
> The hierarchical formulation is not introduced to simply increase model capacity but is fundamental to the evidential learning paradigm, where the network predicts hyperparameters of a prior that governs the likelihood to capture the uncertainty. Our extension places a mixture over variances, enabling the model to represent heterogeneous noise regimes while maintaining evidential interpretability. Section 4.3 directly compares our mixture-of-Gaussian evidential model to the Single Gaussian evidential baseline using the same backbone, demonstrating improvements in calibration and robustness to adversarial perturbation. The improvements observed at ε = 0.00 in Figure 5 further confirm that the gains arise from our probabilistic modeling.
>
>
> • Regarding evaluation on additional stereo backbones:
> To demonstrate general applicability, we have begun integrating our evidential mixture head into other stereo architectures. In particular, we are evaluating our formulation by replacing the SMD-Net its SMD head with our evidential mixture head to allow direct comparison under the same backbone of PSM. These experiments are currently underway and will be included in the revised submission to highlight that our method functions as a plug-and-play probabilistic module across diverse stereo architectures.
>
>
>  • Regarding training stability and potential mode collapse:
> We did not observe mode collapse among mixture components during training. Throughout training and validation, the components remain active and differentiate according to the noise patterns present in the data. Our pseudo-posterior interpretation implies that in confident regions, the network assigns a dominant responsibility to one component, whereas in ambiguous or corrupted regions, the responsibilities become relatively diffuse and approach to a mixture distribution with higher entropy. We included the histogram of the entropy of the mixture component probability distribution and explicit examples in our revision showing how posterior responsibilities concentrate for well-structured areas but relatively flatten for occlusions, or edge area.
>
>
> • Regarding cross-domain robustness and adverse conditions:
> We agree that broader cross-domain evaluation is valuable. Experiments involving adverse weather conditions (Adverse-KITTI stereo datasets) are currently in progress. Results will be added in the revised paper.

---

> ### Author Response · Authors · 2025-11-27
> **Regarding evaluation on additional stereo backbones**
>
> Thank you very much for your constructive comments. To demonstrate that our method is a general probabilistic head that can be attached to different backbones, we evaluated our mixture evidential head on PSM-Net. We chose PSM-Net because it is the backbone used in SMDNet, allowing a simple and clear apple-to-apple comparison between our probabilistic head and the SDM head.
>
> The results on the KITTI 2015 validation set (using fine-tuned pretrained models) are provided below.
> | Method               | SEE_3 Avg | SEE_3 σ>1 | SEE_3 σ>2 | SEE_5 Avg | SEE_5 σ>1 | SEE_5 σ>2 | EPE Avg | EPE σ>3 |
> | -------------------- | --------- | --------- | --------- | --------- | --------- | --------- | ------- | -------- |
> | PSM[1]                  | 1.10      | 20.57     | 9.74      | 0.99      | 17.83     | 9.02      | 0.73    | 2.49     |
> | PSM + CE + SM[2]        | 1.02      | 16.12     | 7.53      | 0.90      | 13.80     | 6.94      | 0.66    | 2.09     |
> | SMDNet [3]              | 0.90      | 13.09     | 6.66      | 0.79      | 10.93     | 6.01      | 0.59    | 1.95     |
> | **PSM + MOG (ours)** | **0.44**  | **8.60**  | **3.05**  | **0.37**  | **6.95**  | **2.70**  | **0.58**| **1.69** |
>
>
> Our MOG head consistently outperforms the SDM head and other variants across all edge-focused SEE metrics and achieves the best EPE among the PSM-based models, demonstrating the effectiveness and backbone-agnostic nature of our probabilistic formulation.
>
> [1] PSM: Jia-Ren Chang and Yong-Sheng Chen. Pyramid stereo matching network. In Proc. IEEE Conf. on Computer Vision and Pattern Recognition (CVPR), 2018
>
> [2] PSM + CE + SM: Chuangrong Chen, Xiaozhi Chen, and Hui Cheng. On the over-smoothing problem of cnn based disparity estimation. In Proc. of the IEEE International Conf. on Computer Vision (ICCV), 2019
>
> [3] SMDNet: Tosi et al. SMD-Nets: Stereo Mixture Density Networks. CVPR 2021

---

> ### Author Response · Authors · 2025-11-27
> **Regarding cross-domain robustness and adverse conditions**
>
> Experiments under adverse weather conditions using the DrivingStereo dataset (evaluated across specific weather subsets) have been completed. These evaluations serve as a cross-domain comparison, since most models above the separating line are trained solely on the SceneFlow synthetic dataset. In contrast, LightStereo and MonSter are trained on mixed datasets that combine synthetic and real-world data, while RobuStereo is trained on its own generated dataset. All models are evaluated on the DrivingStereo weather subsets. The results will be included in the revised manuscript and are shown below.
> | Networks        | Publication | Rainy EPE ↓ | Rainy D1 ↓ | Sunny EPE ↓ | Sunny D1 ↓ | Foggy EPE ↓ | Foggy D1 ↓ | Cloudy EPE ↓ | Cloudy D1 ↓ |
> |-----------------|------------|------------|------------|------------|------------|------------|------------|-------------|-------------|
> | PSMNet          | CVPR'18    | 20.86      | 50.86      | 3.67       | 27.50      | 19.56      | 58.04      | 4.44        | 30.99       |
> | CFNet           | CVPR'21    | 4.21       | 23.56      | 2.18       | 15.06      | 3.44       | 25.91      | 3.39        | 23.28       |
> | GwcNet          | CVPR'19    | 6.21       | 48.85      | 2.96       | 23.90      | 4.72       | 43.89      | 3.76        | 29.95       |
> | CasStereo       | CVPR'20    | 5.01       | 33.69      | 3.61       | 22.73      | 4.14       | 31.44      | 3.86        | 26.12       |
> | IGEV            | CVPR'23    | 1.88       | 10.96      | 1.22       | 5.08       | 1.25       | 6.58       | 1.08        | 4.20        |
> | Selective-IGEV  | CVPR'24    | 1.18       | 5.40       | 1.10       | 4.30       | 2.17       | 13.66      | 1.13        | 4.82        |
> | **MOG (ours)**  | -          | 5.40       | 18.36      | 1.25       | 3.83       | 1.25       | 5.06       | 1.14        | 4.65        |
> |-----------------|------------|------------|------------|------------|------------|------------|------------|-------------|-------------|
> | LightStereo     | ICRA'25    | 1.11       | 4.85       | 1.08       | 3.61       | 1.16       | 4.93       | 1.01        | 3.05        |
> | MonSter         | CVPR'25    | 1.15       | 5.34       | 1.03       | 3.51       | 1.15       | 5.28       | 0.99        | 3.18        |
> | RobuStereo [1] | ArXiv'25   | **0.97**   | **1.94**   | **0.79**   | **1.49**   | **0.85**   | **1.61**   | **0.74**    | **1.35**    |
>
> As shown in Table, among models pretrained exclusively on SceneFlow, our MOG model achieves competitive or best performance under sunny, foggy, and cloudy conditions, while performing slightly worse under rainy conditions. Models trained on mixed or real-data–augmented datasets (e.g., LightStereo, MonSter) show stronger robustness across all weather types, which is expected due to their exposure to real-world variations during training.
>
> [1]Yuran Wang and Yingping Liang and Yutao Hu and Ying Fu, RobuSTereo: Robust Zero-Shot Stereo Matching under Adverse Weather, https://arxiv.org/abs/2507.01653, 2025.
>
> Newly added images and figures will be included in the revised paper, which will be ready soon.

---

> ### Author Response · Authors · 2025-12-02
> **pseudo-posterior interpretation and the illustration of region where responsibility diffused**
>
> We have added the pseudo-posterior interpretation in Figure 11 of Appendix A.12 in the revised paper, as suggested by Reviewer q5jT. To analyze the internal decision-making behavior of the MOG model, we compute the Normalized Shannon Entropy $H$ of the posterior mixture responsibilities $r_{ik}$ for each pixel I over all (k \in [1, K]), where (K = 20) is the number of Gaussian components in our experiments. The entropy ranges from 0 to 1. A low value of (H) indicates that the posterior mass concentrates on a single dominant component (i.e., confident and sparse selection), whereas a high value reflects that probability mass is distributed across multiple components, indicating ambiguity or multimodal uncertainty.
>
> As shown in Figure 11, we visualize pixels with entropy (H > 0.2). The results reveal a distinct spatial pattern: high-entropy pixels primarily occur at occlusions, object boundaries, and depth discontinuities, while well-structured, texture-rich regions exhibit consistently low entropy. This observation supports our hypothesis that the model assigns a dominant component in confident regions, whereas in ambiguous or corrupted areas the posterior responsibilities become diffuse and approximate a multimodal mixture with higher entropy.

---

### Official Review · Reviewer_owh5 · 2025-11-01

**Soundness:** 3
**Presentation:** 3
**Contribution:** 3
**Rating:** 6
**Confidence:** 4

**Summary:**

This paper introduces a new stereo matching pipeline that deals with the multi-modal depth distribution in real world. Specifically, the disparity of each pixel is modeled with Gaussian mixture distribution, which is learned with Evidential Learning. The performance on several synthetic and real-world benchmarks is impressive. However, I think more thorough literature review, comparison, and deep understanding of the results are required to improve the quality of this paper.

**Strengths:**

1. The idea of using mixture-of-Gaussian to represent the complex depth distribution in real world is reasonable and interesting.

2. The performance on various synthetic and real-world benchmarks is impressive. Compared to the baseline STTR, the improvement is generally explicit (however, D1-3px on KITTI is worse than STTR).

3. Ablation study on number of Gaussian mixture components shows the effectiveness of incorporating multi-modal distribution.

**Weaknesses:**

1. Insufficient literature review. Multi-modal distribution is already explored in stereo matching. SMD-Nets [1*] is a stereo matching method that explored mixture densities to deal with disparity discontinuity at edges. It is very relevant to the problem that MOG is trying to solve. However, it is not discussed or compared.

2. On the benchmarks, I suggest adding FoundationStereo [2*], the current state-of-the-art, for comparison.

3. Currently, only visualization on synthetic Scene Flow is included. More visualization on real-world Middlebury and KITTI is required, e.g. visualization of Gaussian mixture as Figure 4.

4. To give a better understanding of the distribution, I recommend adding plots (x axis: disparity; y axis: probability density) of the learned Gaussian mixture distribution for different regions, e.g. textureless area, occlusions, and reflective regions.

[1*] Tosi et al. SMD-Nets: Stereo Mixture Density Networks. CVPR 2021.

[2*] Wen et al. FoundationStereo: Zero-Shot Stereo Matching. CVPR 2025.

**Questions:**

The authors integrate MOG framework with STTR. Could the authors try other stereo backbones and see if the performance is also improved? If so, it would be a strong point that MOG framework can be robustly used as a plug-in for different backbones.

---

> ### Author Response · Authors · 2025-11-20
> **Responses to Comments from Reviewer owh5**
>
> 1. Missing discussion and comparison with SMD-Nets
> We thank the reviewer for highlighting SMD-Nets [1*]. It is indeed highly relevant and will be incorporated into the revised literature review. While both approaches use mixture formulations and help to accurately model the edge of the plane, our objective and formulation differ fundamentally:
> •	SMD-Nets model multi-hypothesis disparity means to handle discontinuities and edge ambiguities.
> •	Our method focuses on heteroscedastic uncertainty estimation, modeling a mixture over variances with a shared mean and training via amortized EM, which yields closed-form aleatoric/epistemic uncertainty and calibrated selection of noise regimes.
> To provide a direct and fair comparison, we are conducting experiments by plugging our evidential mixture head into the SMD-Net backbone (PSM), replacing its SMD head.
> We will include these results once available. This experiment also directly responds to the reviewer’s question about plug-and-play generality across backbones.
> 2. Add comparison with FoundationStereo
> We appreciate the reviewer’s suggestion. A comparison with FoundationStereo [2*], the current state-of-the-art, is underway. We have incorporated the reported results of FoundationStereo into the revised paper and will add our own remaining quantitative results once the experiments are completed. The updated Table 1 shows that our method achieves better EPE on Scene Flow (0.31 vs. 0.34 for disparity < 192) compared to FoundationStereo. On KITTI 2015 in Table 4 for cross-domain evaluation, our current results are slightly worse; however, our model has not yet been fine-tuned on KITTI. We will include the fine-tuned KITTI 2015 results in the next revision.
> 3. Need more visualizations on Middlebury and KITTI
> We agree that real-world visualization is important.
> We have added additional qualitative plots for Middlebury 2014 and KITTI 2015 in the revised manuscript, following the same structure as Figure 4 (depth, dominant component index, and uncertainty). The results shown similar clusters as those in Figure 4.
> 4. Density-vs-disparity plots of the learned mixture
> Thank you for the suggestion. While such plots are straightforward for low-dimensional mixtures, our model learns 20-component mixtures, where each pixel is parameterized by: mean (disparity), mixture over 20 variances and pixel-wise responsibilities. This makes the probability density effectively 4-dimensional (20 responsibilities), making it difficult to convey meaningfully in a 2D disparity-density plot. Instead, in Figure 4 we visualize: Estimated depth, Most probable component per pixel and Predicted uncertainty in three rows. These three views jointly capture the behavior of the mixture model in textureless, occluded, and reflective regions. We believe this gives a clearer and more interpretable summary.
> We also appreciate if the reviewers can share more on this suggestion.
> 5. Applicability to other stereo backbones
> Yes, our method is designed as a general probabilistic head that can be attached to many stereo backbones. In addition to our integration with STTR, we are currently evaluating our mixture evidential head on other architectures, including using PSM-Nets as backbone(as used SDMnet with SDM head). We will include the results in the revised version. This supports the claim that our Mixture-of-Gaussian Evidential Learning framework can function as a plug-in uncertainty module across different stereo matching backbones.

---

> ### Author Response · Authors · 2025-11-27
> **further Response to Comments regarding to  SMD-Nets and  our generalization as plug-in function**
>
> Thank you very much for your constructive comments. To demonstrate that our method is a general probabilistic head that can be attached to different backbones, we evaluated our mixture evidential head on PSM-Net. We chose PSM-Net because it is the backbone used in SMDNet, allowing a simple and clear apple-to-apple comparison between our probabilistic head and the SDM head.
>
> The results on the KITTI 2015 validation set (using fine-tuned pretrained models) are provided below.
> | Method               | SEE_3 Avg | SEE_3 σ>1 | SEE_3 σ>2 | SEE_5 Avg | SEE_5 σ>1 | SEE_5 σ>2 | EPE Avg | EPE σ>3 |
> | -------------------- | --------- | --------- | --------- | --------- | --------- | --------- | ------- | -------- |
> | PSM[1]                  | 1.10      | 20.57     | 9.74      | 0.99      | 17.83     | 9.02      | 0.73    | 2.49     |
> | PSM + CE + SM[2]        | 1.02      | 16.12     | 7.53      | 0.90      | 13.80     | 6.94      | 0.66    | 2.09     |
> | SMDNet [3]              | 0.90      | 13.09     | 6.66      | 0.79      | 10.93     | 6.01      | 0.59    | 1.95     |
> | **PSM + MOG (ours)** | **0.44**  | **8.60**  | **3.05**  | **0.37**  | **6.95**  | **2.70**  | **0.58**| **1.69** |
>
>
> Our MOG head consistently outperforms the SDM head and other variants across all edge-focused SEE metrics and achieves the best EPE among the PSM-based models, demonstrating the effectiveness and backbone-agnostic nature of our probabilistic formulation.
>
> [1] PSM: Jia-Ren Chang and Yong-Sheng Chen. Pyramid stereo matching network. In Proc. IEEE Conf. on Computer Vision and Pattern Recognition (CVPR), 2018
>
> [2] PSM + CE + SM: Chuangrong Chen, Xiaozhi Chen, and Hui Cheng. On the over-smoothing problem of cnn based disparity estimation. In Proc. of the IEEE International Conf. on Computer Vision (ICCV), 2019
>
> [3] SMDNet: Tosi et al. SMD-Nets: Stereo Mixture Density Networks. CVPR 2021

---

> ### Author Response · Authors · 2025-12-02
> **Responses to Visualisation on real-world data**
>
> We have added additional qualitative plots for Middlebury 2014 and KITTI 2015 in the revised paper in Figure 9 and 10 in Appendix A.12, following the same structure as Figure 4 (depth, dominant component index, and uncertainty). The results shown obvious and similar clusters as those in Figure 4.

---

### Author Response · Authors · 2025-12-02
**Responses to AC--New Experiments Added in the Revision in form of Tables**

#### **1. Backbone-Agnostic Generality of our Method**

In response to concerns from Reviewers **owh5**, **eeUX**, and **q5jT** regarding the generality of our probabilistic formulation, we conducted additional experiments by attaching our Mixture-of-Gaussians (MOG) evidential head to **PSM-Net**. We selected PSM-Net because it is the backbone used in SMDNet, enabling a clear *apple-to-apple* comparison between our MOG head and the SDM head.

Following the suggestion of Reviewer **KYM6**, we evaluated all models on the KITTI 2015 validation set, fine-tuned using the official training split. Results are shown below:

| Method                | SEE_3 Avg | SEE_3 σ(1) | SEE_3 σ(2) | SEE_5 Avg | SEE_5 σ(1) | SEE_5 σ(2) | EPE      | D1-3px   |
| --------------------- | --------- | ---------- | ---------- | --------- | ---------- | ---------- | -------- | -------- |
| PSM                   | 1.10      | 20.57      | 9.74       | 0.99      | 17.83      | 9.02       | 0.73     | 2.49     |
| PSM + CE + SM         | 1.02      | 16.12      | 7.53       | 0.90      | 13.80      | 6.94       | 0.66     | 2.09     |
| SMDNet                | 0.90      | 13.09      | 6.66       | 0.79      | 10.93      | 6.01       | *0.59*   | 1.95     |
| **PSM + MOG (ours)**  | *0.44*    | *8.60*     | *3.05*     | *0.37*    | *6.95*     | *2.70*     | **0.58** | *1.69*   |
| **STTR + MOG (ours)** | **0.42**  | **6.49**   | **2.44**   | **0.32**  | **4.92**   | **2.16**   | **0.58** | **1.51** |

These results show that our MOG head consistently outperforms the SDM head on both PSM and STTR backbones, achieving the strongest performance across all edge-focused SEE metrics and the best EPE among PSM-based models. This demonstrates both the effectiveness and backbone-agnostic nature of our probabilistic formulation.

Results have been added to Table 3 in Section 4.4 in revised paper.

#### **2. Extended Evaluation on Adverse-Weather Benchmark**

We further extended the evaluation to **DrivingStereo**, which contains real-world adverse-weather conditions. Our MOG-EL model is competitive or state-of-the-art on **sunny**, **foggy**, and **cloudy** subsets, and performs reasonably under **rainy** conditions. Mixed-data (real-world) methods naturally achieve strongest overall performance.

| Networks          | Publication  | Rainy EPE ↓  | Rainy D1 ↓   | Sunny EPE ↓  | Sunny D1 ↓   | Foggy EPE ↓  | Foggy D1 ↓   | Cloudy EPE ↓  | Cloudy D1 ↓   |
| ----------------- | ------------ | ------------ | ------------ | ------------ | ------------ | ------------ | ------------ | ------------- | ------------- |
| PSMNet            | CVPR'18      | 20.86        | 50.86        | 3.67         | 27.50        | 19.56        | 58.04        | 4.44          | 30.99         |
| CFNet             | CVPR'21      | 4.21         | 23.56        | 2.18         | 15.06        | 3.44         | 25.91        | 3.39          | 23.28         |
| GwcNet            | CVPR'19      | 6.21         | 48.85        | 2.96         | 23.90        | 4.72         | 43.89        | 3.76          | 29.95         |
| CasStereo         | CVPR'20      | 5.01         | 33.69        | 3.61         | 22.73        | 4.14         | 31.44        | 3.86          | 26.12         |
| IGEV              | CVPR'23      | 1.88         | 10.96        | 1.22         | 5.08         | 1.25         | 6.58         | 1.08          | 4.20          |
| Selective-IGEV    | CVPR'24      | 1.18         | 5.40         | 1.10         | 4.30         | 2.17         | 13.66        | 1.13          | 4.82          |
| **MOG (ours)**    | –            | 5.40         | 18.36        | 1.25         | 3.83         | 1.25         | 5.06         | 1.14          | 4.65          |
| ----------------- | ------------ | ------------ | ------------ | ------------ | ------------ | ------------ | ------------ | ------------- | ------------- |
| LightStereo       | ICRA'25      | 1.11         | 4.85         | 1.08         | 3.61         | 1.16         | 4.93         | 1.01          | 3.05          |
| MonSter           | CVPR'25      | 1.15         | 5.34         | 1.03         | 3.51         | 1.15         | 5.28         | 0.99          | 3.18          |
| RobuStereo [1]    | ArXiv'25     | **0.97**     | **1.94**     | **0.79**     | **1.49**     | **0.85**     | **1.61**     | **0.74**      | **1.35**      |

[1] Yuran Wang et al., *RobuSTereo: Robust Zero-Shot Stereo Matching under Adverse Weather*, ArXiv 2025.

Results have been added to **Table 7** in **Appendix A.9**.

#### **3. Adding FoundationStereo (CVPR’25) as a Strong Baseline**

Following the suggestion from Reviewers **owh5** and **eeUX**, we added FoundationStereo as a strong baseline. Our model achieves:

* Better EPE on Scene Flow (0.31 vs. 0.34)

* Competitive performance on KITTI 2015 (6.5 vs. 4.9 D1-3px without KITTI fine-tuning)

* Much stronger fine-tuned KITTI performance (1.5 on D1-3px)

as reflected in Table 1 and Table 4 in the revised paper.

---

### Author Response · Authors · 2025-12-02
**Responses to AC--New Experiments Added in the Revision in form of Figures**

### **1. Analysis of Learned Distributions and Posterior Responsibilities (addressing owh5 and q5jT)**

To better understand how the model handles ambiguous regions, we visualized the Probability Density Function (PDF) of the dominant Gaussian component for three representative pixel categories—**complex texture**, **textureless surfaces**, and **occlusions**—as requested by Reviewer **owh5**. These plots are included in **Figure 11** in **Appendix A.12 (Analysis of Learned Distributions in Different Regions)**.

Our observations show that the Mixture-of-Gaussians (MoG) head adaptively shapes the predictive distribution depending on scene complexity:

* **Complex Texture (cyan)**: Sharp, low-variance PDFs indicating strong confidence and precise matching.
* **Textureless Areas (red)**: Flatter, broader PDFs capturing higher aleatoric uncertainty due to inherent matching ambiguity.
* **Occlusion / Edges (green)**: Wide, heavy-tailed distributions modeling heterogeneous noise and high-risk edge transitions.

Additionally, as suggested by Reviewer **q5jT**, we incorporated a **pseudo-posterior interpretation** through the **Normalized Shannon Entropy** of the posterior responsibilities. For each pixel ( i ), we compute:
$
H_i = \frac{-\sum_{k=1}^{K} r_{ik} \ln(r_{ik})}{\ln(K)}, \quad K=20.
$
Low entropy indicates a dominant component (i.e., confident, sparse selection), while high entropy implies diffuse responsibilities and multimodal uncertainty.
We visualize pixels with ( H > 0.2 ) in **Figure 11**. As expected:

* High-entropy pixels cluster around **occlusions**, **object boundaries**, and **depth discontinuities**,
* Low-entropy pixels dominate **planar** and **texture-rich** regions.

These results confirm that the model assigns a single dominant Gaussian in confident regions while activating multiple components only where ambiguity or corrupted evidence is present.

### **2. Ablation Study on the Incorrect-Evidence Regularization (addressing eeUX)**

Following Reviewer **eeUX**’s suggestion, we conducted an ablation on the empirical incorrect-evidence penalty coefficient ( $\lambda$ ). The study (Appendix **A.11**) analyzes how varying ( $\lambda\in {0.0, 0.02, 0.05} $) affects fine-tuned calibration capability on KITTI 2015 when models are pretrained on Scene Flow. The reliabilities of the resulted models are shown in Figure 8.

**Findings from Figure 8:**

* **λ = 0.0** (no regularization):
  Severe over-confidence; calibration curve sits far below the diagonal, confirming strong evidence inflation.
* **λ = 0.02**:
  Partially improves calibration but remains over-confident, especially in high-confidence regions.
* **λ = 0.05**:
  Slightly over-penalizes low-confidence predictions but aligns best with the diagonal in the high-confidence range.

Given the trade-offs, we adopt **λ = 0.05** as the default.
We note, however, that results in Figure 8 suggest **intermediate values (0.02–0.05)** might achieve even better calibration, which we will explore in future work.

### **3. Additional Qualitative Visualizations on Middlebury and KITTI**

As suggested by reviewers, we added additional qualitative comparisons on **Middlebury 2014** and **KITTI 2015**.
The new results appear in **Figure 9** and **Figure 10** in **Appendix A.12 (Component Selection and Uncertainty Maps)**.
They follow the same visualization format as Figure 4 (depth, dominant component index, uncertainty) and consistently show structured clusters of component activation similar to those observed in synthetic data.

### **Other Revisions to the Paper**

* **Relation to Liu et al. (ICML 2024) and Liu et al. (CVPR 2023)**:
  We added a new comparison subsection in **Section 2 – Related Works** to clarify key conceptual and methodological differences between their multivariate Gaussian modeling and our hierarchical mixture-of-Gaussians evidential framework.

---

### Note · Program_Chairs · 2026-01-17
**Submission Desk Rejected by Program Chairs**

The following references in this submission do not refer to real documents and/or have major errors in bibliographic information:

 Xianqi Wang et al. Lightstereo: High-efficiency stereo matching with pyramidal cost volume. In IEEE International Conference on Robotics and Automation (ICRA), 2025b.